# Throwing and manipulating and cheating with a DNA nano-dice

Xiaochen Tang[1,2,4], Tianshu Chen[1,2,4], Wenxing Li[3,4], Dongsheng Mao[3], Chenbin Liu[3], Qi Wu[3], Nan Huang[3], Song Hu[3], Fenyong Sun [3] ✉, Qiuhui Pan [1,2] ✉ & Xiaoli Zhu [3] ✉

Artificial molecular machines have captured the imagination of researchers, given their clear potential to mimic and influence human life. Key to behavior simulation is to reproduce the specific properties of physical or abstract systems. Dice throwing, as a stochastic model, is commonly used for result judgment or plan decision in real life. In this perspective we utilize DNA cube framework for the design of a dice device at the nanoscale to reproduce probabilistic events in different situations: equal probability, high probability, and low probability. We first discuss the randomness of DNA cube, or dice, adsorbing on graphene oxide, or table, and then explore a series of events that change the probability through the way in which the energy released from entropy-driven strand displacement reactions or changes in intermolecular forces. As such, the DNA nano-dice system provides guideline and possibilities for the design, engineering, and quantification of behavioral probability simulation, a currently emerging area of molecular simulation research.

In the 1980s, Sauvage et al. connected two cyclic molecules into a chain and named it a catenane which took the first step towards molecular machines[1]. Similar to the assembly of macroscopic machines by machine components, molecular machines assemble various molecular units together and operate under external stimuli such as electrical, chemical or light energy, thereby realizing the overall operation of molecular machines[2,3]. After about 40 years of development, researchers have designed and manufactured a large number of molecular machine components that can be assembled on the nanoscale like Lego bricks, including molecular switches, molecular ratchets, molecular motors, molecular linkages, molecular rings, and so on[4–6]. Therefore, the construction of artificial intelligence molecular machines has a good connection with nanoscale molecular devices and simulated biological motors, which would draw a blueprint in the fields of biology and chemistry.

DNA, owing to its highly parallel structure that can be accurately predicted and assembled, has been recognized as an ideal smart nanomaterial, and has been widely used in controllable molecular computing tools, construction of complex three-dimensional structures, and functional nanomachines[7–9]. In addition, DNA can be modified in various ways and folded in many forms so as to enable the synthesized molecular devices to perform more complex operations, rather than based on simple movable linking fragments (e.g. sigma single bond, pi-metal complex and cyclic molecules)[10]. Hence, it has become an important research field and hot spot to design and fabricate DNA nanomachines in recent years. For example, Qian group first created single-stranded DNA robots to carry out the task of picking up target cargos and delivering them to specified destinations at the nanoscale, which achieved a major breakthrough in the field of molecular robotics and brought many possibilities for the application of simulating behavior or operation in a macroscopic system[11].

Rolling dice is a common behavior in real life, which is usually used for result judgment or plan decision in various fields[12]. When Albert Einstein lettered to Max Born in 1926 "At any rate, I am convinced that *He* (God) does not play dice", it became a sign of his

[1]Department of Clinical Laboratory Medicine, Shanghai Children's Medical Center, School of Medicine, Shanghai Jiao Tong University, Shanghai 200127, P. R. China. [2]Shanghai Key Laboratory of Clinical Molecular Diagnostics for Pediatrics, Shanghai 200127, P. R. China. [3]Department of Clinical Laboratory Medicine, Shanghai Tenth People's Hospital of Tongji University, Shanghai 200072, P. R. China. [4]These authors contributed equally: Xiaochen Tang, Tianshu Chen, Wenxing Li. ✉e-mail: sunfenyong@263.net; panqiuhui_med@163.com; xiaolizhu@shu.edu.cn

opposition to quantum mechanics and its randomness, but he took it for granted that dice throwing is a random process. With the discovery of deterministic chaos, it was conceivable that dice throwing might be described by perfectly deterministic laws. Owing to the speciality of dice throwing, it can be used as a stochastic model, and also represent events with different probability by changing conditions to perform certain logical operations. Here, we would like to utilize DNA framework for the design of a dice device at the nanoscale to imitate different situations of dice throwing.

In this work, as a realization of this concept, we construct a DNA cube-based system for the simulation of playing dice. This artificial nano-dice system mainly consists of two parts: regular cube framework formed by DNA self-assembly as dice and graphene oxide (GO) as throwing platform. Four single-stranded DNA (ssDNA) oligonucleotides like tentacles sticking out from four opposite vertices of the cube can interact with the GO through π-π stacking, thereby binding the DNA nano-dice to the GO, i.e. the throwing table. The ssDNA is labeled with fluorophore, which could fluoresce on the opposite side of the GO surface or be quenched while plastered to GO, so that the synthetic DNA system could be explored the possibilities of mimicking the process of throwing dice according to whether the fluorescence on the ssDNA of DNA nano-dice is quenched or not. In the present work, we first verify the randomness of throwing process. By using entropy-driven strand displacement reactions, we next try to manipulate the dice directionally to achieve artificial flipping. Finally, according to the different affinity of bases to GO and the multiple intermolecular forces between DNA and GO, we change the base composition of ssDNA on nano-dice or chemically modified the ssDNA to cause events with unequal probability in order to reach the goal of cheating. In general, we demonstrate these facts with a simple and intuitive model, which is a caricature of dice (DNA) on the table (GO), showing that there is broader development of DNA nanotechnology in the field of simulation and more potential applications based on this model.

## Results

### Design of DNA nano-dice system

To simulate a dice, we selected several DNA oligonucleotides to design a wireframe DNA cube as an addressable nano-dice, whose side length is of 20 nt (~7 nm) (Fig. 1A)[13]. The DNA scaffold is composed of four 88 nucleotide (nt) DNA stands (s1 to s4), which hybridize with one another to yield a regular cube. Then we used four 20 nt DNA stands (x) and four 34 nt DNA stands (a to d) to hybridize eight single-stranded segments at the top and bottom faces, thereby forming a DNA cube with ssDNA of 12 nt (black arrow) at each of four symmetrical corners. As shown in Fig. 1B, these 12-nt-long tentacle-like ssDNAs (named as t-DNA) are of unique sequences, and labeled with fluorophore (FAM) to represent four different signals (F1 to F4), respectively. In experiments, we performed fluorescence detection four times in parallel, that is, t-DNA-1 to t-DNA-4 of the dice was labeled with FAM in turn and only one was modified, while left other three unmodified. According to every two signals meaning one face of DNA cube, four signals can express six faces based on the law of permutation and combination so that each face of the cube can correspond to different pips from "1" to "6", like "1" (pip) representing by F1 and F2. Therefore, we regard this kind of DNA cube as a nano-dice for subsequent behavior simulation.

Next, we applied GO as a platform for dice throwing because GO can adsorb the single-stranded t-DNA of the DNA cube to mimic the dice throwing process. In addition, as an excellent acceptor of fluorescence resonance energy transfer (FRET), GO can quench the fluorescence of the fluorophore-labeled t-DNA adsorbed on the surface for qualitative or quantitative detection. Thus, through detecting whether the fluorescence of the t-DNA on DNA nano-dice is quenched or not, we can correspondingly know the number of pips thrown by the dice. We named the face adsorbed on the GO as "D" and its opposite face as "U", so when GO quenches F3 and F4, the "U" face would show "1" (pip), and we describe this situation as "$U_{F1,2}$; $D_{F3,4}$" (Fig. 1C). As a result, we reasoned that the six situations of DNA cube adsorbing on GO could be explained by "U" and "D".

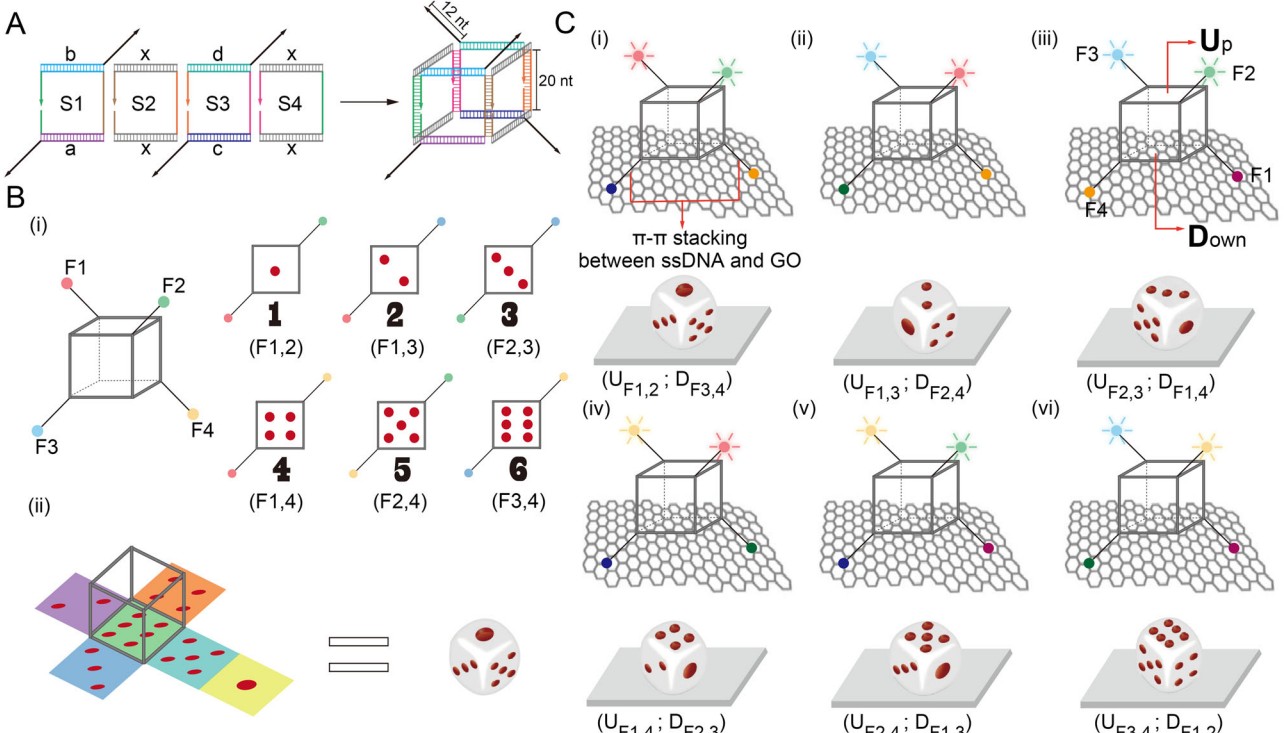

**Fig. 1 | Design of DNA cube-based system for mimicking dice throwing.**
**A** Schematic illustration of DNA cube. **B** Dice simulation at the nanoscale using DNA cube with FAM-labeled t-DNA at four symmetrical corners. **C** The design of DNA nano-dice adsorbed on GO and presentation of six throwing results. U: top face of dice. D Bottom face of dice.

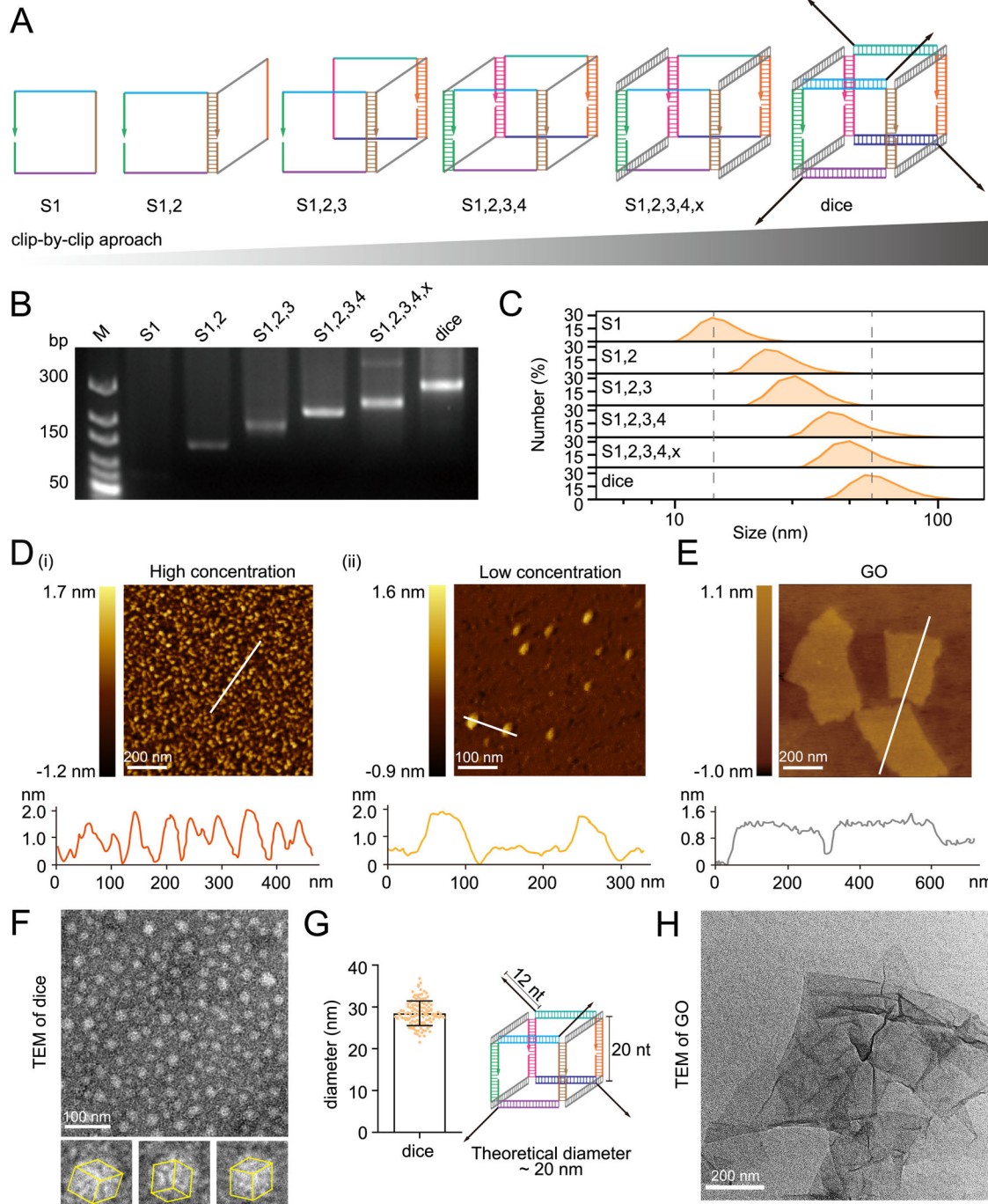

**Fig. 2 | Assembly and characterization of the DNA nano-dice system.**
**A** Construction process of DNA nano-dice. **B** AGE characterization of DNA nano-dice from S1 to dice from 3 independent experiments. **C** DLS of nano-dice from S1 to dice. **D** AFM characterization of DNA nano-dice in high and low concentration from 3 independent experiments. The scale bars are 200 and 100 nm, respectively. **E** AFM image of GO sheets deposited on mica substrates from 3 independent experiments. The scale bar is 200 nm. **F** Negative-staining TEM for characterization of DNA nano-dice. The scale bars are 100 nm and 20 nm, respectively. **G** Quantification of diameter analysis for the nano-dice in (**F**), $n = 122$ independent dice spots. Data are presented as mean values ± s.d. **H** TEM image of GO sheets with homogeneous monolayer region from 3 independent experiments. The scale bar is 200 nm. Source data are provided as a Source Data file.

### Construction and characterization of DNA nano-dice system

We first investigated the various stages of the DNA nano-dice assembly process (Fig. 2A). The assembly outcome of different phase products was characterized by using multiple experimental techniques. With the stepwise addition of each DNA strand, we verified if DNA nano-dice could be successfully assembled by agarose gel electrophoresis (AGE) (Fig. 2B). Results show a clear shift of the bands towards the larger fragment as the assembly progresses. Moreover, dynamic light

scattering (DLS) also confirmed the gradual increase of hydration radius (Fig. 2C). These results together indicate that the nano-dice can assemble step-by-step as expected. In addition, atomic force microscopy (AFM) and transmission electron microscope (TEM) were used to characterize the structure of the nano-dice vividly. The cube-like nanostructures were observed as a dot in the AFM imaging (Fig. 2D), verifying the successful formation of DNA nano-dice[14]. Parallelly, GO was characterized by AFM, showing a typical layer appearance and a

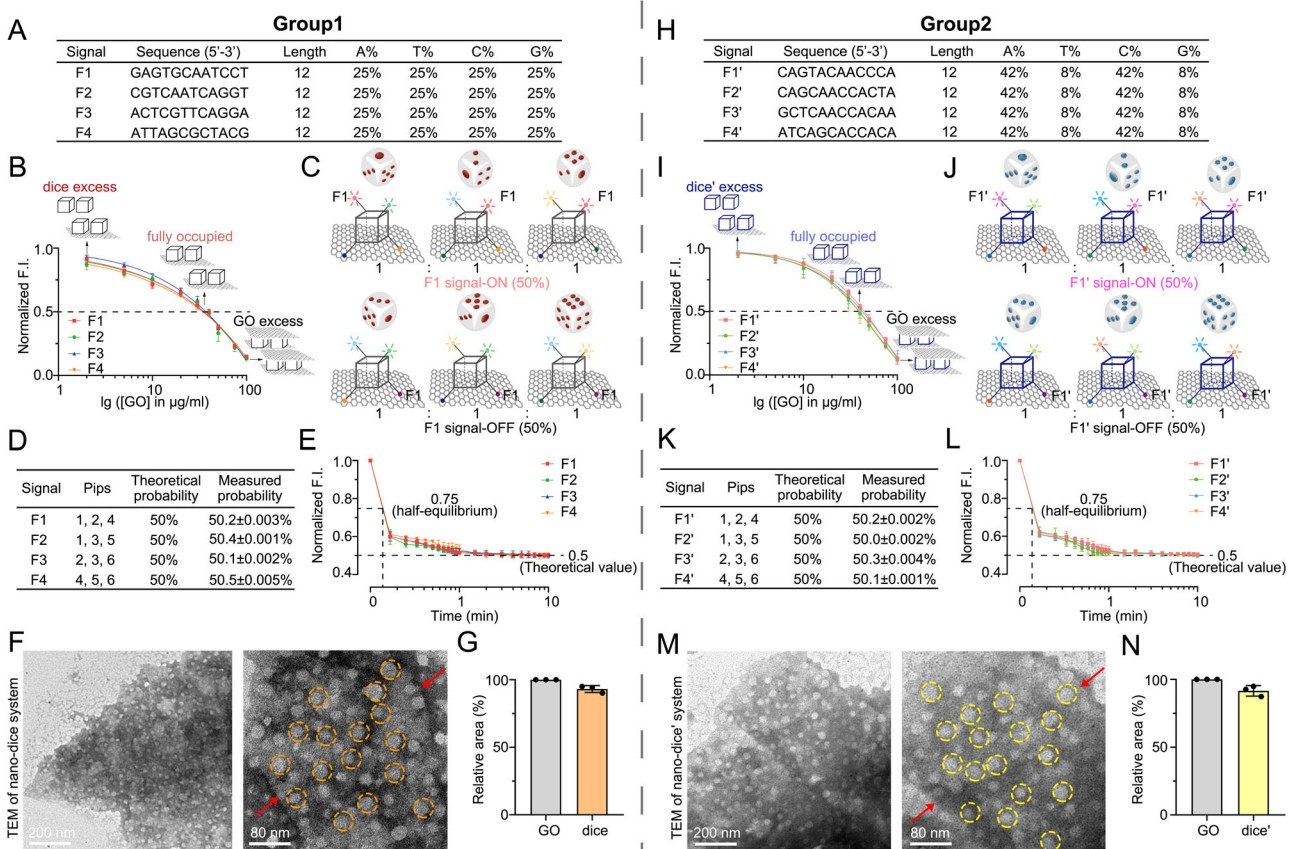

**Fig. 3 | The stochastic effect performance of DNA nano-dice system. A** Sequence information table of four signals in the nano-dice. **B** Concentration-dependent fluorescence changes of four signals after the nano-dice was mixed with the GO, *n* = 3 independent experiments. **C** Schematic illustration of the stochastic effect of the nano-dice system. **D** The result table of stochastic effect of the nano-dice system. **E** Time-dependent fluorescence changes of four signals after the nano-dice was mixed with the GO (41.45 μg/ml), *n* = 3 independent experiments. **F** Negative-staining TEM for characterization of DNA nano-dice system. Red arrow: creases of GO. Orange circle: DNA nano-dice (average diameter: 28.6 ± 2.9 nm). The scale bars are 200 nm and 80 nm, respectively. **G** Quantitative analysis of the relative area of DNA nano-dice in GO (92.7 ± 3.4%), *n* = 3 independent samples. **H** Sequence

information table of four signals in the nano-dice'. **I** Concentration-dependent fluorescence changes of four signals after the nano-dice' was mixed with the GO, *n* = 3 independent experiments. **J** Schematic illustration of the stochastic effect of the nano-dice' system. **K** The result table of stochastic effect of the nano-dice' system. **L** Time-dependent fluorescence changes of four signals after the nano-dice' was mixed with the GO (47.59 μg/ml), *n* = 3 independent experiments. **M** Negative-staining TEM for characterization of DNA nano-dice' system. Red arrow: creases of GO. Yellow circle: DNA nano-dice' (average diameter: 27.7 ± 3.5 nm). The scale bars are 200 nm and 80 nm, respectively. **N** Quantitative analysis of the relative area of DNA nano-dice' in GO (91.6 ± 3.9 %), *n* = 3 independent samples. Data are presented as mean values ± s.d. Source data are provided as a Source Data file.

thickness of about 1.5 nm (Fig. 2E). Negative-staining TEM imaging similarly confirmed the geometry of DNA nano-dice, and counted the average diameter (28.5 ± 2.9 nm) of particles (Fig. 2F, G). Also, GO appeared on TEM images as homogeneous and featureless regions (Fig. 2H). The above results together convincingly proved that the two components of the DNA cube-based system, namely the "dice" and the "table", were successfully constructed.

### Stochastic effect of DNA nano-dice system

Generally, as the number of dice throws increases, the number of occurrences of "1" to "6" pips should be closer and closer because each pip has the same probability of 1/6. To demonstrate the feasibility of using DNA nano-dice system instead of thousands of throwing, we first analyzed the equivalence of the t-DNA at each of four symmetrical corners. Due to the adsorption of fluorophore-labeled oligonucleotides on GO surface depends on DNA length and base difference[15], we strictly constrained the sequence composition of the four t-DNAs on the nano-dice to make them equivalent when binding GO. As shown in Fig. 3A and Supplementary Fig. 1, the sequences though are disordered, the proportions of A/T/C/G and the length are fixed at 25% and 12 nt in group 1, respectively. For each fluorophore (F) at the

terminal of the t-DNA, its probability of being in the Up and Down positions on GO is 50% each (Fig. 1C), and the former is signal ON while the latter is GO-quenched signal OFF. Therefore, when the fluorescence is quenched to half of the original, it can be equivalent to the embodiment of randomness. On this assumption, we optimized the work ratio of GO and DNA dice in order to achieve the purpose of randomly throwing dice with equal probability events. In experiments, we fixed the concentration of DNA dice (500 nM), and adjusted the dice/GO ratio by changing the concentration of GO. Figure 3B and Supplementary Fig. 2 show that the fluorescence is gradually quenched with the increase of GO concentration. Through the concentration curve, when 50% of the fluorescence is quenched, we get an optimal dice/GO ratio (500 nM vs. 41.45 μg/ml, namely $1.2 \times 10^{-5}$ mol/g, junction of theoretical value dashed line and concentration curve in Fig. 3B), under which the dice would exactly and fully occupy the GO surface. When the GO concentration is relatively low (dice/GO ratio >$1.2 \times 10^{-5}$ mol/g, above the theoretical value dashed line in Fig. 3B), there are many free and un-adsorbed DNA nano-dice in the mixture, indicating that the number of rolling is not enough to verify randomness. On the contrary, when dice is relatively low (dice/GO ratio <$1.2 \times 10^{-5}$ mol/g, below the theoretical value dashed line in Fig. 3B), GO

would cover multiple faces of a dice and result in the total quenching of all fluorophores at four symmetrical corners of the dice, indicating that the throwing platform exceeding dice will affect the probability assessment. Both of these two modes have errors in understanding when compared with actual dice throwing situation, so probability simulation can be perfectly fitted at the optimal dice/GO ratio ($1.2 \times 10^{-5}$ mol/g). In addition, we calculated the theoretical GO/dice ratio when dice completely occupied the GO surface based on the theoretical area of GO and dice through the following equation:

$$S_{GO} \times C_{GO} = S_{one\,dice} \times (C_{dice} \times NA) \tag{1}$$

where $S_{GO}$ is the surface area of GO ($2.63 \times 10^{-3}$ m$^2$/µg)[16], $C_{GO}$ is the theoretical concentration of GO, $S_{one\,dice}$ is the theoretical bottom area of DNA nano-dice (max: ~$39.8 \times 10^{-17}$ m$^2$), $C_{dice}$ is the concentration of DNA nano-dice (500 nM), NA is the Avogadro's constant ($6.02 \times 10^{23}$ /mol), and $(C_{dice} \times NA)$ is the number of all DNA nano-dice ($3.01 \times 10^{14}$ mol/ml).

Therefore, the results of theoretical GO concentration we obtained were 45.5 µg/ml, which is in good consistence with the above measurements.

Under the optimized ratio, the randomness of rolling dice should be represented by the equivalence of the four fluorescent signals (F1, F2, F3 and F4), each with a 50% quenching rate. Taking signal F1 as an example, F1 would have a 50% chance to release or quench the signal at the optimal GO concentration theoretically (Fig. 3C). When F1 signal is "ON", it means that "1"/ "2"/ "4" pips have been rolled. On the contrary, "3"/ "5"/ "6" pips mean F1 signal "OFF". In experiments, we have separately detected the four fluorescence signals of the nano-dice in the presence of GO (Fig. 3D). As expected, the results showed that all measured probabilities were closely around the theoretical value (50%) by using the following formula:

$$P_{pip} = 1/6 \times E_{Fx,y} \tag{2}$$

where 1/6 means the theoretical random probability of each pip, and $E_{Fx,y}$ means average of normalized Fx and Fy (x,y = 1, 2, 3, 4).

Therefore, we can obtain the probability of each pip is consistent with the theoretical probability (1/6), suggesting the successful verification of stochastic effect in this model system. Besides, the speed of GO quenching is observed to be very fast and it could reach a steady state after only about one minute (Fig. 3E). The almost identical changing trends of the four fluorescent signals further illustrated the equivalence of the four t-DNAs and the randomness of rolling dice. More figuratively, the DNA nano-dice system was characterized by negative-staining TEM (Fig. 3F), and we could further calculate the total area of GO and dice by image-j software respectively to define the occupancy between the two, showing that the dice would be almost fully occupied in GO ($92.7 \pm 3.4\%$) (Fig. 3G).

Furthermore, we constructed another nano-dice by changing the sequence composition of the four 12-nt t-DNAs, in which A/C accounts for 42% and T/G accounts for 8% in group 2 (Fig. 3H and Supplementary Fig. 1). We named this DNA cube as nano-dice'. Similar to nano-dice (group 1), dice' (group 2) could also mimic the randomness of dice throwing with success (Fig. 3I–L), and the dice'/GO ratio required was 500 nM vs. 47.59 µg/ml ($1.05 \times 10^{-5}$ mol/g) at this time (Fig. 3I). The difference in concentration may be caused by the adsorption capacity of GO and bases. In addition, we employed negative-staining TEM for the characterization of the DNA nano-dice' system (Fig. 3M, N), which was consistent with the characterization results of the DNA nano-dice system (dice' accounted for $91.6 \pm 3.9\%$ of GO). The above results show that although nano-dice and nano-dice' have different A/T/C/G composition ratios in the four corners, their own four sides of A/T/C/G composition ratio are the same, thereby the constructed dice having the randomness of throwing. Based on this observation, it is tempting

to speculate that we can construct various types of nano-dice to reproduce the stochastic effect.

## Artificial manipulation of DNA nano-dice system

Next, to simulate the behavior of changing the pips of dice by applying external force in reality, we devised some strategies based on DNA hybridization reaction and strand displacement reaction (SDR) in the DNA nano-dice system (see below). It is known that the ssDNA adsorbed on GO can be desorbed by the addition of its complementary DNA[17], which can compete with GO to bind the adsorbed ssDNA. Upon forming dsDNA, the DNA bases are hidden in the helical structure and only the negatively charged phosphate groups are exposed, thereby disrupting the hydrophobic interactions between ssDNA and GO and leading to the desorption of the ssDNA from GO[18,19]. In our DNA nano-dice system, a short complementary single-stranded DNA of 15 nt (named as c-DNA) was used to induce the desorption of t-DNA from GO through base pairing. Triggered by the c-DNA, we hypothesize that the t-DNA forming double strands would leave the GO surface, and another t-DNA on other corners of the same dice would be further adsorbed, turning the dice to another pip.

As planned, we first monitored the situation of adding one c-DNA. Taking c-DNA-3 (complementary to t-DNA-3) for example, upon addition of the c-DNA-3, F3 signal would be restored so that the original representative pips ("1", "4", "5") with quenched F3 signal would change to pips ("2", "3", "6") and the original representative pips ("2", "3", "6") with unquenched F3 signal would remain. To verify the above speculation, the fluorescent signal of F before and after the addition of c-DNA will be measured to indicate the flip of the nano-dice. But before that, to make the ratio of the fluorescence changing before and after meaningful, two other DNA cubes adsorbed on GO were constructed as positive and negative controls, respectively (Supplementary Fig. 3A). For positive control (named as pc-dice), the cube consists of a cubic framework and two instead of four t-DNAs sticking out from two opposite vertices at one face, whose opposite side is labeled with a fluorophore directly at one vertices. In the case of negative control (named as nc-dice), the cube also consists of a cubic framework and two t-DNAs sticking out from two opposite vertices at one face. But the fluorophore is positioned differently, it is labeled at the terminal of one of the t-DNAs (Supplementary Fig. 3A, B). In the absence of the c-DNA, we define the fluorescence value measured by the pc-dice with GO as 1 and the fluorescence value measured by the nc-dice as 0 (Supplementary Fig. 3C), since the fluorophore on pc-dice is theoretically always away from GO, whereas the fluorophore on nc-dice is always close to GO. Similarly, we investigated the rolling situations of different dice at the whole level using confocal laser scanning microscopy (CLSM) and at the single-molecule level using total internal reflection fluorescence microscopy (TIRF) (Fig. 4). CLSM allowed us to characterize the overall fluorescence change on a GO slice. When the nc-dice rolling on GO, there was almost no fluorescence visible (Fig. 4A, left), while the fluorescence of normal dice and pc-dice nearly occupied half and all of GO area, respectively (Fig. 4B, C, left). The statistics of average fluorescence density also corroborates the observation results (Fig. 4D). Characterization via TIRF revealed different situations in three kinds of dice rolling, demonstrating the distribution of dice on GO and randomness of normal dice rolling (Fig. 4A–C, right). Each bright spot can be seen as a dice, and different distributions of spots were showed in the respective TIRF images of F1 to F4 signals. Through the spot counts in the limited range (300 nm × 300 nm), it showed that the number of bright spots in pc-dice was about twice that of normal dice, so were the four signals (Fig. 4E). Moreover, the results of CLSM and TIRF images in normal dice also confirmed stochastic effect of DNA nano-dice system in Fig. 3, consistent with the fluorescence spectrum detection.

On the basis of defining the upper and lower limits above and confirming that c-DNA can hybridize to nano-dice (Supplementary

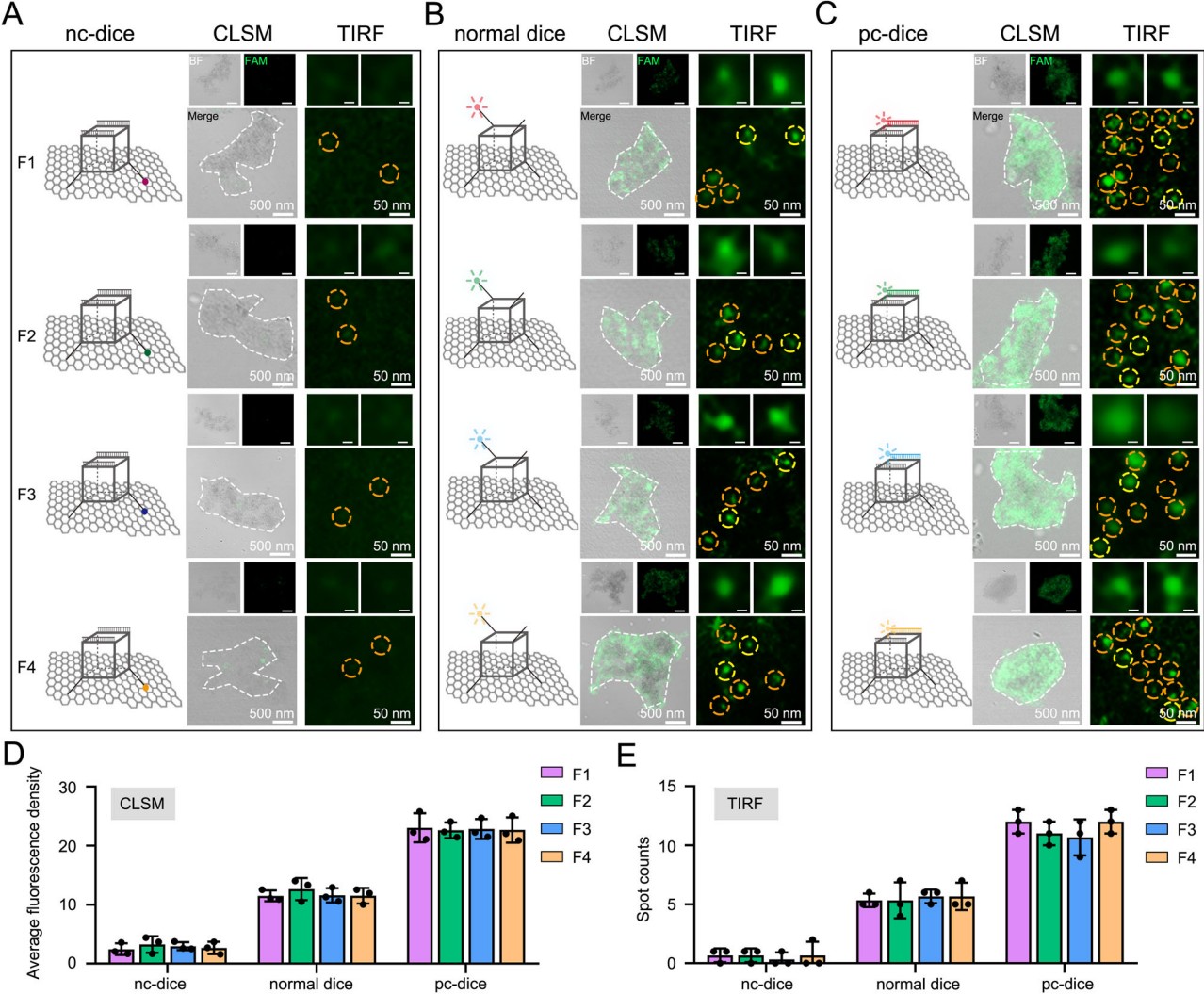

**Fig. 4 | The fluorescence imaging characterization of DNA nano-dice system.**
**A** Confocal images (left) and single-molecule images (right) of four signals in nc-dice with GO. The scale bars are 500 nm in split and merge CLSM images (left), 50 nm in original TIRF images and 10 nm in enlarged TIRF images (right), respectively. **B** Confocal images (left) and single-molecule images (right) of four signals in normal dice with GO. The bright spots in yellow circle were enlarged to show. The scale bars are 500 nm in split and merge CLSM images (left), 50 nm in original TIRF images and 10 nm in enlarged TIRF images (right), respectively. **C** Confocal images (left) and single-molecule images (right) of four signals in pc-dice with GO. The bright spots in yellow circle were enlarged to show. The scale bars are 500 nm in split and merge CLSM images (left), 50 nm in original TIRF images and 10 nm in enlarged TIRF images (right), respectively. **D** Quantitative fluorescence density analysis of four signals for each confocal sample in different groups, $n = 3$ independent experiments. **E** Statistical analysis of spot counts for each TIRF sample in different groups in 300 nm × 300 nm, $n = 3$ independent experiments. Data are presented as mean values ± s.d. Source data are provided as a Source Data file.

Fig. 4), we investigated the fluorescence signal changes of the nano-dice after adding c-DNA-3. The relative fluorescence signal of each F is calculated by the following formula:

$$R_F(\%) = (F_i - F_{nc})/(F_{pc} - F_{nc}) \times 100\% \qquad (3)$$

where $F_i$, $F_{nc}$ and $F_{pc}$ are the measured fluorescence intensity of the nano-dice, pc-dice and nc-dice, respectively.

In the flipping model, we can predict that nano-dice would only appear three pips ("2", "3", "6") after adding c-DNA-3 (Fig. 5A). It should be noted that the F3 signal are all at the "U" face, and other three signals account for 1/3 upon this condition, that is theoretically the relative fluorescence intensity of F3 should be 1 (100%), and the rest should be 1/3 (33.3%), respectively. As is shown in Fig. 5B, F3 fluorescence signal gradually increased and the final fluorescence efficiency reached a value (~90%) close to that of the theoretical positive control (100%, pc-dice) after about 90 min, while the other three fluorescence signals (F1, F2 and F4) progressively decrease until ~30%, a value close

to the theoretical 33.3%. As a control, the addition of ssDNA non-complementary to t-DNA (non-c-DNA) was also examined and led to an increase in all fluorescence signals (Supplementary Fig. 5). According to the law of mass action, when the GO surface is almost saturated with adsorbed DNA, the incoming non-c-DNA molecules simply displace the pre-adsorbed DNA molecules on the GO surface and this effect is equivalent to each signal.

Considering from the perspective of dice flipping, the complete desorption of t-DNA-3 and further adsorption of one of the other three t-DNA indicated the change of dice pips, meaning that the appearance of six pips ("1" to "6"; the probability of each pip is 1/6) turned to three pips ("2", "3", "6"; the probability of each pip is 1/3). We also adopted the other three c-DNAs, which is corresponding to t-DNA-1, t-DNA-2 and t-DNA-4, respectively, to investigate the changes of the pips of the nano-dice. Analogous to the result of the nano-dice in the presence of c-DNA-3, the fluorescent signal of t-DNA-x (x = 1, 2, 4) would be restored and the other three signals would continue to decrease to ~30% when the c-DNA-x (x = 1, 2, 4) corresponding to t-DNA-x was

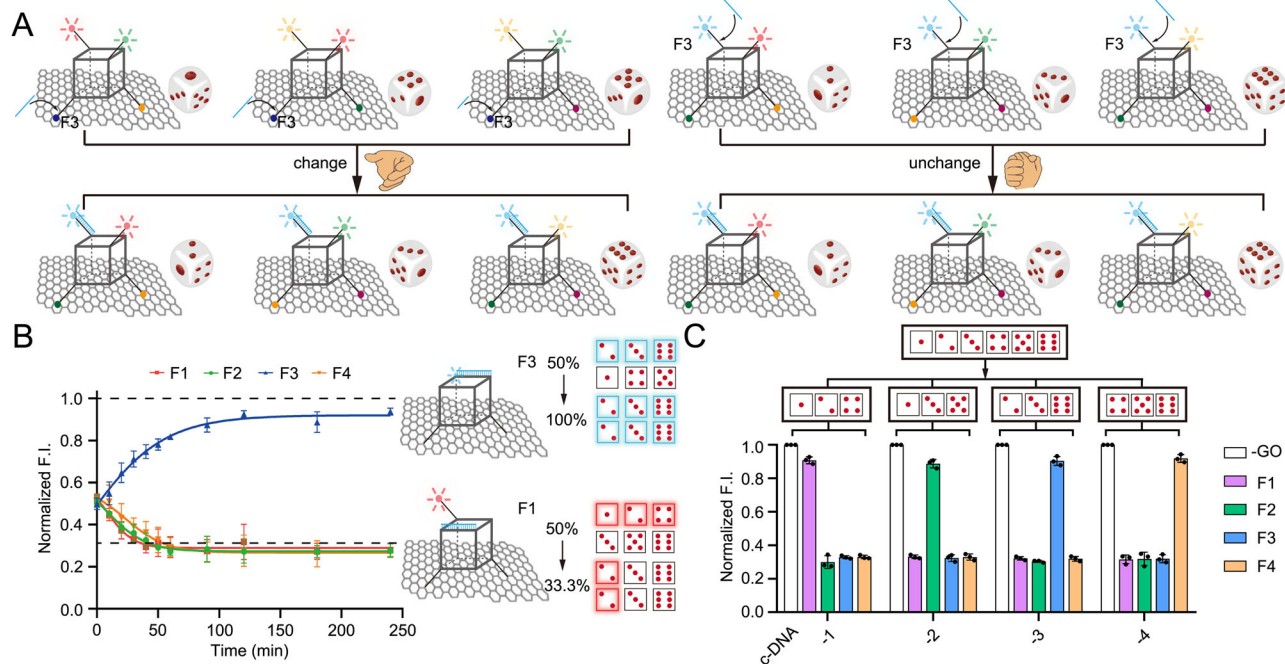

**Fig. 5 | Artificial manipulation of DNA nano-dice system by single regulator.**
**A** Schematic illustration of c-DNA-3 adding to the DNA nano-dice system. **B** Kinetics of four signals change induced by adding the c-DNA-3, *n* = 3 independent experiments. Right, schematic illustration of theoretical changes in the pips and fluorescence of dice. **C** Normalized fluorescence intensity of four signals after adding four kinds of c-DNA, separately, *n* = 3 independent experiments. Data are presented as mean values ± s.d. Source data are provided as a Source Data file.

added (Fig. 5C and Supplementary Fig. 4, Supplementary Table 2), elucidating the successful manipulation of the DNA nano-dice and the equivalence of the four t-DNAs in the manipulation.

Furthermore, we examined how two c-DNA together influences the change of DNA nano-dice. c-DNA-3 together with c-DNA-4 was added to the nano-dice, and the kinetic curve of four signals was monitored. Theoretically, c-DNA-3 and c-DNA-4 would hybridize with t-DNA-3 and t-DNA-4 respectively to release them from the surface of GO and thereby enhance F3 and F4 signals. Thereafter, the only possibility is that t-DNA-1 and t-DNA-2 adsorb on the surface of GO, that is, all the dice flips to the face of "6" (pip) (Fig. 6A and Supplementary Fig. 6). The observation was consistent with the above hypothesis that complete recovery of F4 and F3 fluorescence signals (50% to 100%) was achieved and the other two fluorescence signals were thoroughly quenched (50% to 0%) in sufficient time (Fig. 6B). Similarly, we combined two c-DNAs with each other and added them to the mixture of nano-dice and GO, showing the situation that the two fluorescence signals increased and the other two fluorescence signals decreased to represent the number of pips flipped to a unique one (Fig. 6C and Supplementary Fig. 6, Supplementary Table 3), that is, random events are manipulated into unique events. Under the manipulation of dual regulators, we can also observe the fluorescence (F1 to F4) of six pips at the single-molecule level. Exemplary CLSM and TIRF images shown in Supplementary Fig. 7 indicated that at each pip, the two fluorescence signals would be quenched, and the other two fluorescence signals would be completely presented, corresponding to their respective pips (Fig. 1C). The overall change trend of fluorescence was also in accord with the spectrum data.

To revert the manipulated nano-dice back to its original random state, it can be simply achieved by adopting a complementary oligonucleotide of c-DNA (named as cc-DNA) to rival t-DNA and thus release the c-DNA from the t-DNA. The sequence of cc-DNA is similar to t-DNA with the only difference that cc-DNA has an additional 3-nt toehold at the 3′-terminal. Thus, it could competitively hybridize with c-DNA with a higher affinity than c-DNA and t-DNA through a process known as

SDRs. The competitive binding relationship among t-DNA, c-DNA and cc-DNA was first verified by PAGE (Supplementary Fig. 8). Then cc-DNA-1 was added to the DNA nano-dice system as a deregulator, to deactivate unequal events caused by c-DNA-1 (Fig. 6D). Fluorescence monitoring of kinetics revealed that all four signals were up- or down-regulated to the same trend (~50%) and tended to a steady state (Fig. 6E, F, Supplementary Table 4), demonstrating the success of this strategy. We replicated other fluorescence change by adding corresponding cc-DNA-x (x = 2, 3, 4) on the basis of the nano-dice with c-DNA-x (x = 2, 3, 4), respectively. In accord with the changes caused by cc-DNA-1, four fluorescent signals would all adjusted to ~50% (Supplementary Figs. 9–11, Supplementary Table 4), implying that unequal events successfully reversed to random equal events.

We further mimic the manipulation of dice from one specific pip to another at the nanoscale by combinatorial application of different c-DNA and cc-DNA couples. In the above, we adopted c-DNA-3 and c-DNA-4 to fix the evenly distributed pips to a specific pip "6" (Fig. 6C). On this basis, c-DNA-2 and cc-DNA-3 were further added, the former is used to dissociate the t-DNA-2 from GO, while the latter is used to unblock the t-DNA-3 by c-DNA-3 (Fig. 6G). In this way, the pip of all the nano-dice is expected to flip from "6" to "5", which can be represented by the changes of the fluorescence of F2 and F3 in the experiment. As is shown in Fig. 6H and Supplementary Fig. 12, an increase (10% to 100%) in fluorescence intensity of F2 signal and a decrease (100% to 10%) in fluorescence intensity of F3 signal are observed, while the F1 and F4 signals are almost unaffected during this dynamic change. In view of the complexity of the whole system, the time to reach steady fluorescence was prolonged, but we still visualize the correct fluorescence changes, confirming successful flipping behavior driven by entropy change in catalytic reaction of short-strand nucleic acid hybridization.

From the above, we can conclude that efficient and rational control of the DNA nano-dice can be achieved through the artificial manipulation of the adsorption and desorption between GO and t-DNA. By introducing a regulator, i.e. the c-DNA, to compete with GO to bind corresponding t-DNA, the probability of the six sides of the

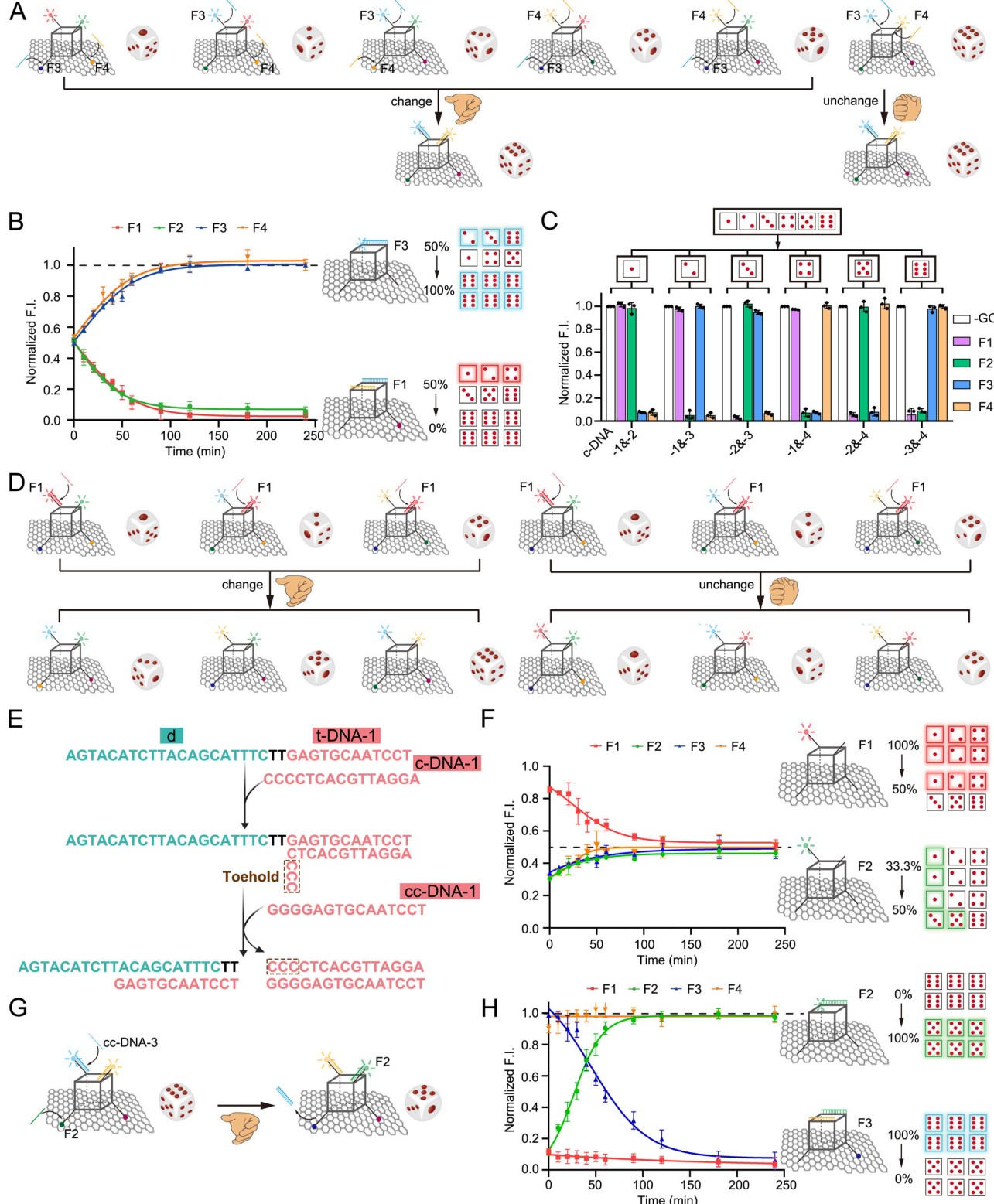

**Fig. 6 | Artificial manipulation of DNA nano-dice system by dual regulators.**
**A** Schematic illustration of c-DNA-3 and c-DNA-4 adding to DNA nano-dice system.
**B** Kinetics of four signals change induced by adding the c-DNA-3 and c-DNA-4, $n = 3$ independent experiments. Right, schematic illustration of theoretical changes in the pips and fluorescence of dice. **C** Normalized fluorescence intensity of four signals after adding two kinds of c-DNA at the same time, $n = 3$ independent experiments. **D** Schematic illustration of cc-DNA-1 adding to DNA nano-dcie system with c-DNA-1. **E** Schematic illustration of SDR among t-DNA-1, c-DNA-1, and cc-DNA-

1. **F** Kinetics of four signals change induced by adding the cc-DNA-1, $n = 3$ independent experiments. Right, schematic illustration of theoretical changes in the pips and fluorescence of dice. **G** Schematic illustration of c-DNA-2 and cc-DNA-3 adding to DNA nano-dice system with c-DNA-3 and c-DNA-4. **H** Kinetics of four signals change induced by adding the c-DNA-2 and cc-DNA-3, $n = 3$ independent experiments. Right, schematic illustration of theoretical changes in the pips and fluorescence of dice. Data are presented as mean values ± s.d. Source data are provided as a Source Data file.

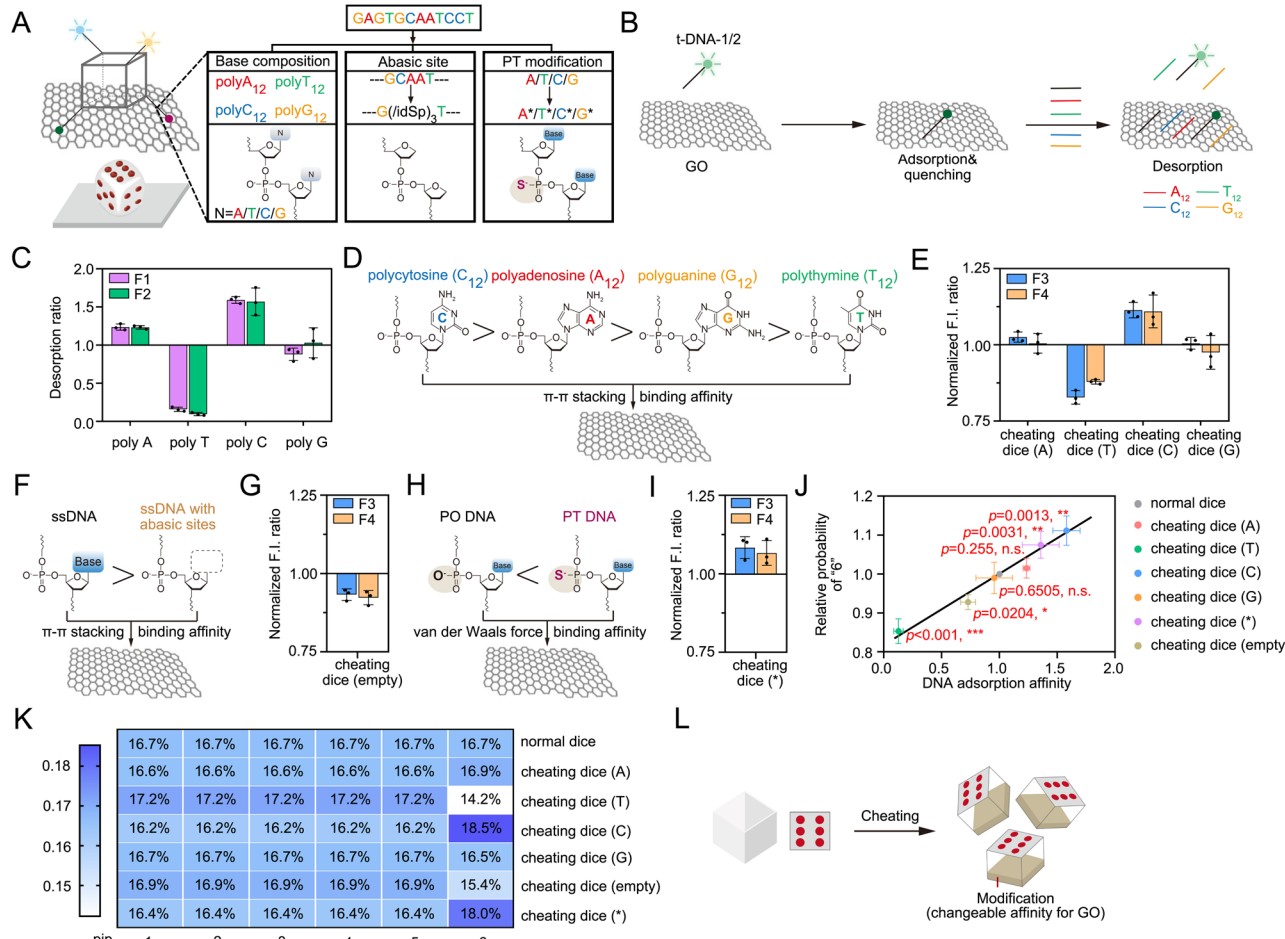

**Fig. 7 | The cheating behavior of DNA nano-dice system. A** Schematic illustration of three cheating methods for nano-dice. PT modification: phosphorothioate modification. **B** Schematic illustration of adsorption and desorption of original t-DNA-1/2 by adding poly $A_{12}/T_{12}/C_{12}/G_{12}$. **C** Desorption ratio of FAM labeled original DNA (group 1) from GO by poly $A_{12}/T_{12}/C_{12}/G_{12}$ after 60 min reaction, $n = 3$ independent experiments. Data are presented as mean values ± s.d. **D** Schematic illustration of the differences of the binding affinity between poly$N_{12}$ (N = A, T, C or G) and GO. **E** Normalized fluorescence intensity ratio of cheating dice (A/T/C/G), $n = 3$ independent experiments. Data are presented as mean values ± s.d. **F** Schematic illustration of the differences between normal t-DNA and t-DNA with abasic sites on GO. **G** Normalized fluorescence intensity ratio of cheating dice (empty), $n = 3$ independent experiments. Data are presented as mean values ± s.d. **H** Schematic illustration of the differences between PO DNA and PT DNA on GO. PO DNA: DNA with phosphodiester backbone; PT DNA: DNA with phosphorothioate modification.

**I** Normalized fluorescence intensity ratio of cheating dice (*), $n = 3$ independent experiments. Data are presented as mean values ± s.d. **J** Correlation analysis between DNA adsorption affinity and relative probability of "6", $n = 3$ independent experiments. X-axis: average of the two bars in t-DNAs (F1, F2) for each cheating methods, indicative of DNA adsorption strength. Y-axis: average of the two bars in nano-dice (F3, F4) for each cheating methods, reflecting the relative probability of throwing "6" (pip). The cheating effect of each cheating dice compared with the normal dice was verified by one-way analysis of variance. *$p$ value < 0.05, **$p$ value < 0.01, ***$p$ value < 0.001. Data are presented as mean values ± s.d. **K** The absolute probability of "1" to "6" (pips) with different cheating dice. Formula: $P_{pip6} = 1/6 \times E_{3,4}$ and $P_{pip1/2/3/4/5} = (1- P_{pip6})/5$ (the remaining 5 pips are of random equal probability). **L** Schematic illustration of cheating dice for throwing "6" (pip). Source data are provided as a Source Data file.

nano-dice adsorbed on GO will no longer be random. Some pips will not appear anymore, while others will be more likely to occur. Further introducing of a deregulator, i.e. the cc-DNA, to remove the control from the c-DNA, the nano-dice can also revert to its original random state. Moreover, complicated manipulation of different pips with equal or unequal probability can be achieved by combinatorial application of different c-DNA and cc-DNA couples. It is also possible to develop various types of manipulation, e.g. aptamer, which would make the nano-dice smartly responsive.

### Cheating behavior of DNA nano-dice system
As the card sharp always cheating in gambling games, we want to reproduce this cheating behavior (e.g. rolling the largest number "6") on the nanoscale based on the special adsorption characteristics between GO and DNA. It is known that the binding of ssDNA to the GO surface involves different types of interactions such as hydrogen

bonding[20], hydrophobic[21], and π−π stacking[22] by the theoretical and experimental methods. Here, by altering base composition or by chemical modification to affect the binding affinity of ssDNA to GO, we developed three cheating strategies (Fig. 7A).

Firstly, we investigated the dice cheating by modulating base composition. Here, in order to understand the respective roles of A/T/C/G in the base composition, we directly used poly$N_{12}$ (N = A, T, C or G) to interact with GO. In a competition experiment, we found that poly$A_{12}$, poly$T_{12}$, poly$C_{12}$ and poly$G_{12}$ exhibited competitive differentiation for t-DNA-1/2, whose sequence though is different the A/T/C/G composition is equal, i.e. 25% each (Fig. 7B). As is shown in Fig. 7C and Table S5, adding poly$C_{12}$ induced the highest desorption of t-DNA-1/2 from GO, indicating that poly$C_{12}$ has the highest affinity with GO. A gradient of binding capacity can be also obtained: poly$C_{12}$ > poly$A_{12}$ > poly$G_{12}$ > poly$T_{12}$ (Fig. 7C, D). In the nano-dice, by replacing t-DNA-1 and t-DNA-2 with poly$N_{12}$ (N = A, T, C or G),

respectively, we got a "cheating" dice with a changed probability of pip "6", represented by the changes in the fluorescence signal of F3 and F4 (Fig. 7E and Supplementary Fig. 13). Consistent with the competitive experimental results above, here cheating dice(C) produced the greatest signal, suggesting that polyC$_{12}$ has a much stronger affinity on GO and the probability of pip "6" appearing gets the biggest improvement. Through calculation from the Eq. 2, we obtained the probability of pip "6" appearing as: 18.5%, 16.9%, 16.7%, 16.5% and 14.2% for cheating dice (C), cheating dice (A), original nano-dice, cheating dice (G), and cheating dice (T), respectively. Also, analogous to using dice (group 1) as control to check, we investigated the sequences in dice' (group 2), and the results were basically consistent with group 1 (Supplementary Fig. 14). Thus, further by fine-tuning the sequence of t-DNA, such as replacing one or several bases to C or T, it is able to finely increase or decrease the probability of a certain pip to a theoretical range of 14.2%–18.5%, which is defined by using polyT$_{12}$ and polyC$_{12}$.

Secondly, with regard to DNA sequence-dependent adsorption, it is generally accepted that DNA bases are aromatic and can stack with GO through π-π stacking[23]. Thus, we studied whether the presence of abasic sites on the DNA might be available for dice cheating. Here, we set three consecutive abasic sites in the middle of the t-DNA-1/2. Before testing the performance of the cheating dice, we first measured the adsorption capacity of the altered DNA on GO through a competition experiment similar to that described above. As expected, compared to t-DNA-1/2, the adsorption capacity between altered DNA and GO was reduced to 70 ± 2% of the original (Supplementary Fig. 15A, B, Supplementary Table 5), which should be attributed to the base deletion reduces the binding affinity between ssDNA and GO (Fig. 7F). In the nano-dice model, we then inset three consecutive abasic sites in the middle of both t-DNA-1 and t-DNA-2 to construct a cheating dice (named as cheating dice (empty)) (Supplementary Fig. 15C). When the cheating dice (empty) was thrown onto GO, we found that the fluorescence signal of F3 and F4 representing the pip "6" were reduced by about 10% (Fig. 7G and Supplementary Table 6), indicating the decreased probability of throwing "6" (pip) from original 16.7% to about 15.4%. Similarly, by inserting different numbers of abasic sites on other t-DNA, it is also possible that the probability of the occurrence of the corresponding pips will be reduced to varying degrees, or even the pip will not appear at all.

Thirdly, chemical modification of DNA was employed for dice cheating. Recent investigations showed that phosphorothioate modified DNA (PT DNA), a unique epigenetic modification with a non-bridging oxygen atom replaced by a sulfur atom on the sugar-phosphate backbone, adsorbed more tightly on GO than unmodified DNA with the same sequence[24], which was also verified in our competition experiment (Supplementary Fig. 16A, B, Supplementary Table 5). By using the PT modified t-DNA-1/2 in the nano-dice, the third type of cheating dice (named as cheating dice (*)) is constructed (Supplementary Fig. 16C). With the enhanced binding affinity of the PT modified t-DNA-1/2, the fluorescence of F3 and F4 representing pip "6" also increased (Fig. 7H, I and Supplementary Table 6), suggesting the probability of pip "6" was enhanced to 18.0%. In addition to the PT modification, there are also various alternative strategies for chemical modification of DNA either on its nucleobases and/or the phosphate backbone[25]. For example, the interaction between peptide nucleic acids (PNA) and GO is reported to be more effective due to the elimination of charge repulsion by neutral skeleton on PNA[19]. These various chemical modifications would expand dice cheating strategies and performance.

Combining the data of the cheating dice above, comparison of different cheating strategies on the probability of the occurrence of pip "6" is obtained and displayed in Fig. 7J, K, where the former shows the relative change of the probability, and the latter shows the absolute probability of occurrence. From Fig. 7J, it is interesting to observe a linear relationship between the bind affinity and the relative

probability of pip "6". This result strongly suggests that binding affinity is the key to dice cheating, independent of the specific strategies that alter affinity. This is very similar to dice cheating in reality by adding different substances such as lead to specific positions inside the dice, thereby changing the center of gravity of the dice to achieve the purpose of cheating (Fig. 7L). From Fig. 7J, K, it is also concluded that compared with cheating dice (empty) and cheating dice (*), the cheating strategy by changing the base composition has the highest cheating efficiency, among which cheating dice (C) and cheating dice (T) can maximize and minimize the probability of pip "6", respectively. It should be pointed out that for cheating dice (empty), since we only put three abasic sites in the 12 nt t-DNA, its true upper and lower bounds have not yet been obtained. In summary, different strategies to modulate the binding affinity of t-DNA to GO by altering t-DNA were successfully developed to realize dice cheating. Similarly, tampering with GO might also be a potential cheating way.

## Discussion

Supramolecular chemistry has created quite sophisticated artificial molecular machines in the past decades[26,27], but these machines are still difficult to compete with natural molecular machines, such as DNA, which can be built into DNA nanorobots with dynamic functions such as stepping, structure opening and closing, target capture, etc[11,28,29]. However, the research on DNA nanomachines is still in its infancy, and there are many challenges in tracking, navigation, sustainable precise manipulation and other aspects. Here, we have demonstrated a DNA nano-dice for throwing, manipulating and cheating by using the adsorption characteristics between DNA and GO. We successfully simulated random equal probability events ($P_{pip} = 1/6$) of dice throwing, and controlled nano-dice efficiently to realize the unequal probability events by introducing regulators and deregulators. Also, rather than throwing dice at random, we may ask for getting the maximum pips ("6"). We tested different cheating strategies to increase or decrease the probability of throwing "6", in which cheating dice (C) can maximize the probability from 16.7 to 18.5% and cheating dice (T) can minimize the probability from 16.7 to 14.2%, respectively.

Benefiting from the programmability and predictability of DNA nanotechnology, framework nucleic acids with a cubic structure is available to mimic a dice not only in shape but also in the pips representation. And as graph structures, such DNA cubes on GO are equivalent to dice on the table in reality. Even though the designed system is entirely synthetic, it still can get the corresponding probability by converting the change of the fluorescent signal, namely, converting the result in the nano-scale range into the throwing probability in real behavior. Perhaps the goal to generate stochasticity on DNA-based nanoscale system is not restricted to our nano-dice system. For example, after determining the start location and destination, a cargo-sorting DNA robot could be activated to generate reversible strand displacement reaction through the trigger strand so as to realize random walking on the track[11]. However, this random walking on linear tracks would mean that they rely on the competition and displacement between DNA strands. In other words, the main advantageous feature of our DNA nano-dice system in simulating randomness is to use the existing force between DNA and GO to analyze and compare, with nothing extra, which can be directly interfaced with behavior processes in reality.

Because we wanted to control the rolling of the nano-dice not only for random simulation but also for precise manipulation, we decided to use regulator DNA strands for the change in rolling probability. With effort, the following aspects of the nano-dice could be further manipulated. First, the nano-dice could flip to specific pips by adding single/dual regulators, which could help us understand how to adjust probability. Second, deregulators were introduced to revert the manipulated nano-dice back to its original random state. From our

understanding of how the interaction force between DNA and GO affects the adsorption, we believe that it is possible to regulate the flipping behavior by using sequences with stronger binding energy to compete with nano-dice[30,31]. Third, driven by entropy change in catalytic reaction of short-strand nucleic acid hybridization, the nano-dice could be manipulated from one specific pip to another purposefully. Aptamers, small chemicals and metal nanoparticles might also be used in the nano-dice system for the smart response[32–34]. Thus, DNA nano-dice could work together with the regulators/deregulators to allow the transform of devices with clear purposes from components that are originally randomly distributed. Finally, the working environment of DNA nano-system may be further extended to the programmable biological diagnosis[35,36]. For example, abnormal regulated microRNAs can be used as regulators to trigger signals to indicate the occurrence of diseases.

We have also demonstrated various cheating strategies of the nano-dice, which had a linear relationship between the bind affinity and the relative probability of pip "6", regardless of specific strategies. Our experiment confirmed that we could obtain the maximum/minimum possibility of throwing pip "6" by changing the sequence base to $polyC_{12}$ and $polyT_{12}$. Of course, considering the existing results, the probability does not seem to be a very large change, which may be due to the limited length (12 nt) in the t-DNA of dice, resulting in a limited space for modification. Theoretically, in addition to the above cheating methods that have been proven to be controllable, other strategies might also be used to adjust the throwing probability. For example, peptide nucleic acids (PNAs) are nondegradable DNA mimics in which negatively charged deoxyribose phosphate backbone is replaced by a neutral N-(2-aminoethyl) glycine one[37]. A PNA-GO-based platform was developed for the sensitive and selective detection of DNA, which the PNA-DNA duplex might have additional hydrogen-bonding interactions through amides and π-π stacking interactions with the surface of GO[19]. $_L$-polycarbamate nucleic acids ($_L$-PCNA)[38] and enantiomeric $_D$-PCNA[39] were further synthesized and available with higher binding affinity to DNA, which had more efficient desorption from the GO surface upon addition of the complementary DNA. In the cheating strategy triggered by PT modification, since DNA can interact with surfaces via nucleobases and the phosphate backbone, the addition of soft metals (e.g., Au, $Cu^{2+}$, and $Cd^{2+}$) are expected to increase the interaction, while hard metals (e.g., $Mg^{2+}$) might weaken the interaction, so that it has a certain impact on the adsorption of dice and GO, leading to rolling bias[40]. However, these methods either require the synthesis of nucleic acids with special properties or the addition of other small molecules, which may lead to an overly complex reaction environment. On the other hand, considering the simulation of real cheating events, excessive cheating is easy to be seen through, so nano-dice cheated by using existing strategies can achieve the goal perfectly. Similarly, cheating to the extreme can also be understood as artificial manipulation (that is, directly flipping the dice to the desired pip), which we have manipulated the dice to the desired pip by adding dual regulators successfully. Surely, it is believed that relevant researches can be further optimized in the future, so as to make the cheating more significant and hidden more perfectly.

More generally, our interest is in developing simple nanodevices and controllable building blocks, which the strategy of controlling the DNA nano-dice system is expected to be applied for intelligent molecular nano-controllers and nano-operators. For instance, in the context of the DNA nano-dice system, fluorescence could be selectively changed through the target recognition, and thus specific output 'pips' could be activated. This could be used as a sensor output or a molecular actuator for triggering downstream biomolecular processes. With simple interaction between nano-dice and GO, they could perform even more sophisticated tasks, like logic circuits or color coding. With systematic approaches, the nano-dice system could be easily programmed like molecular macroscopic dice rolling, but working in nanoscale microscopic environments.

## Methods

### Materials and reagents
All oligonucleotides used in this work (Table S1) were synthesized with high-performance liquid chromatography (HPLC) purification by hippo Biotech. Co., Ltd. (Zhejiang, China), which DNA with adasic sites was synthesized by Sangon Biotech. Co., Ltd. (Shanghai, China). Graphene oxide was obtained from XFNANO Materials Tech Co., LTD. (Nanjing, China) without any modification. All solutions were prepared with Milli-Q water (18.2 MΩ·cm) from a Milli-Q purification system (Millipore).

### Assembly of DNA nano-dice
The equal molar ratio of customized single-stranded oligonucleotide strands (500 nM) were mixed in 80 μL of buffer (20 mM Tris, 2 mM EDTA, 12.5 mM $MgCl_2$, pH 7.4) to form the DNA nano-dice. The mixture was first heated at 95 °C for 5 min, 80 °C for 3 min and cooled slowly to room temperature. After the assembly of DNA nano-dice, the nanostructures were purified with Amicon centrifugal filter (10 kDa molecular weight cut-off) and then centrifuged at 10,000 × g for 10 min three times to remove unhybridized ssDNA.

### Toehold-mediated strand displacement reaction
For in vitro verification of the strand displacement reaction system, reaction chains (1 μM) were mixed in buffer (20 mM Tris, 2 mM EDTA, 12.5 mM $MgCl_2$, pH 7.4) and then incubated at 37 °C for 1 h to complete the reaction. For the manipulation process of DNA nano-dice, DNA nano-dice (500 nM) and cc-DNA (1.5 μM) for displacement were mixed and incubated at 37 °C for 4 h for continuous monitoring of kinetics.

### Electrophoresis analysis
For the characterization of DNA nano-dice, we used 2.5% agarose gel electrophoresis with 1× TAE at 80 V in an ice-water bath for 30 min. For the verification of the strand displacement reaction products, we used 20% nondenaturing polyacrylamide gel electrophoresis in 1× TBE at 120 V in an ice-water bath for 90 min. Subsequently, the gel was analyzed using a Gel Doc XR Imaging System (Bio-Rad).

### Characterization of DNA nano-dice and GO
For atomic force microscope (AFM) characterization, DNA nano-dice (25 nM and 500 nM) was dropped on a freshly cleaved mica surface (Yunfeng Co. Ltd., China). After 2 min, the sample was mixed with 50 μL 1× TAE-$Mg^{2+}$ buffer and 2 μL 100 mM $Ni^{2+}$. The graphene oxide sheets (12.5 μg/mL) were also deposited on the surface of mica substrate, and dried with protection from light. Subsequently, all samples were imaged on Agilent 5500 AFM (Agilent Technologies). For negatice-staining transmission electron microscopy (TEM), DNA nano-dice, GO, and the complex system were stained using a 2% aqueous uranyl formate solution and characterized using Hitachi TEM system at 120 kV.

### DLS characterization
Samples of DNA nano-dice (500 nM) and other mid products (500 nM) were diluted 100 times with water for dynamic light scattering analysis. Then they were determined by dynamic laser scattering (DLS) at 25 °C using a commercial laser light scattering instrument (Zetasizer Nano ZSE ZEN3700, Malvern Instruments Ltd.).

### Fluorescence analysis
The fluorescence spectrum was determined using a F-7000 fluorospectrophotometer (HITACHI, Japan). The dye of single-stranded DNA (ssDNA) on each side of DNA nano-dice (group 1) is FAM, which was was excited at 488 nm, and the fluorescence emission spectra was recorded from 500 to 650 nm. For each detection, the ssDNA of nano-

dice was only modified one FAM fluorophore, and the fluorescence detection was performed four times in parallel (F1 to F4). That is, when detecting F1 signal, only t-DNA-1 of DNA nano-dice was labeled with FAM, and t-DNA-2 to t-DNA-4 were not modified. The dye of DNA nano-dice' (group 2) is FAM, which was was excited at 496 nm, and the fluorescence emission spectra was recorded from 500 to 650 nm.

Then, GO at a series of concentrations was mixed with DNA nano-dice for 10 min at room temperature to determine the work concentration of GO.

For the DNA nano-dice manipulation experiment, short ssDNA (named as c-DNA) was incubated with corresponding ssDNA (named as t-DNA) on DNA nano-dice by complementary base pairing at room temperature for 4 h to observe kinetic changes. The concentration of DNA nano-dice used was 500 nM (DNA nano-dice: c-DNA = 1: 3) in buffer (20 mM Tris, 2 mM EDTA, 12.5 mM $MgCl_2$, pH 7.4). After 2 h of c-DNA incubating with DNA nano-dice, DNA complementary to the c-DNA (named as cc-DNA) was incubated with the above mixture at room temperature and observed kinetic changes for 4 h to complete toehold-mediated strand displacement reaction. The concentration of DNA nano-dice used was 500 nM (DNA nano-dice: c-DNA: cc-DNA = 1: 3: 3) in buffer (20 mM Tris, 2 mM EDTA, 12.5 mM $MgCl_2$, pH 7.4). For the measurement of DNA nano-dice with different ways of manipulation, the results were determined by F-7000 fluorospectrophotometer.

For the detection of DNA nano-dice cheating, we have identified three different ways. Three kinds of modified single-stranded DNA (1.5 μM) were firstly added into the mixture (GO with original DNA) (modified DNA: original DNA = 3: 1) to define the adsorption affinity of modified DNA. DNA cheating dice (500 nM) were then mixed with GO for 10 min at room temperature and measured by F-7000 fluorospectrophotometer.

## Fluorescence imaging characterization of the DNA nano-dice system

The samples were detected by LSM 900 Zeiss confocal microscope (Zeiss, Germany) and TIRF (OLYMPUS IX83, Ex:488 nm) at the whole and single-molecule level to measure the change of F1 to F4 signals in cases of random rolling and artificial manipulation. For the detection of random rolling, GO (41.45 μg/ml) was mixed with nc-dice, normal nano-dice, and pc-dice (500 nM) for 10 min at room temperature, respectively. The diluted samples (10 nM, 10 μl) were dropped in the glass slide, and then measured by LSM 900 and OLYMPUS IX83 for the whole and single-molecule characterization. For the detection of artificial manipulation, the concentration of DNA nano-dice used was 500 nM (DNA nano-dice: c-DNA = 1: 3) in GO (41.45 μg/ml). After 2 h of two c-DNAs incubating with DNA nano-dice, the measurements were detected by OLYMPUS IX83.

## Statistics and reproducibility

All numerical data, AFM imaging data, fluorescence imaging data are collected from a minimum of three independent experiments unless otherwise specified. No data were excluded in the studies. Numerical data are presented as mean values ± s.d. Fluorescence spectral data were analyzed with the GraphPad Prism 8.0.1. TEM imaging data were processed by image-j software. TIRF imaging data were analyzed with CellSens software. Statistical mean and differences were evaluated using Microsoft excel 2021' s statistical tools and the GraphPad Prism 8.0.1.

## Reporting summary

Further information on research design is available in the Nature Portfolio Reporting Summary linked to this article.

## Data availability

All data needed to evaluate the conclusions in the paper are present in the paper and/or the Supplementary Information file. Source data are provided with this paper. Additional data related to this paper may be requested from the authors. Source data are provided with this paper.

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

## Acknowledgements

This work was supported by the National Natural Science Foundation of China (Grant Nos. 32150011 and 22074090 to X.Z., 81930066 to F.S., 81871727 to Q.P.), the major project in the basic research field of Shanghai Science and Technology Innovation Action Plan (22JC1402300 to Q.P.), Program of Shanghai Academic Research Leader (18XD1402600 to Q.P.) and Pudong New Area Municipal Health Commission of Shanghai (PW2019D-10 to Q.P.).

## Author contributions

Conceptualization, X.Z., Q.P., and F.S.; methodology, X.Z., X.T., T.C., and W.L.; investigation, X.T., T.C., D.M., and Q.W.; visualization, W.L., D.M., C.L., N.H., and S.H.; acquisition, Q.P. and F.S.; supervision, X.Z., Q.P., and F.S.; writing-original draft, X.T., T.C., and W.L.; writing-review&editing, X.Z., X.T., T.C., Q.P., and F.S. All authors contributed to the final text of the paper.

## Competing interests

The authors declare no competing interests.
