## [Peer Review File · Nature Communications]

Throwing and manipulating and cheating with a DNA nano-diceREVIEWER COMMENTS

Reviewer #1 (Remarks to the Author):

This manuscript describes the use of a DNA cube as a "die" that can be "thrown" by incubating it with graphene oxide sheets; the cubes are adsorbed onto the sheets and the directionality is determined by the quenching (or otherwise) of fluorescent dyes conjugated to the cube via single-stranded DNA oligos. This allows the distribution of die orientations relative to the graphene oxide to be calculated. Furthermore, the manuscript shows that various strategies can be used to bias the die, or "cheat", including by adding oligonucleotides complementary to the dye-carrying protruding strands and by modulating the backbone chemistry of those strands. Both of these approaches essentially weaken the interactions between those strands and the graphene oxide; these need to be balanced carefully to make a die that is "fair".

The paper is fairly well written, although it is odd that the text continually refers to the structure as a hexahedron rather than a cube: hexahedron is a more general term and it seems that the more specific one would be more appropriate here.

The data demonstrating the formation of the cubes is somewhat hard to follow; the figures are rather small and hard to make out (especially the AFM imagery). Also I found it somewhat surprising that all of the data on the behaviors of the dice are in the form of bulk fluorescence readings; given that the "roll" of each individual die is (or at least should be) an independent stochastic event, I would expect to see at least some fluorescence microscopy data here to if nothing else validate that the bulk fluorescence values are at least representative of what one would see if one were to inspect the individual nano-dice one at a time and count up the observed scores from each.

I also wonder what does it mean to "change the concentration of graphene oxide"? The main text is not especially clear on the details of this protocol but it seems to me that graphene oxide sheets are incubated in the solution freely diffusing with the dice; this is inferred from the discussion that you can get two sheets binding to a single die if there is an excess of graphene oxide with respect to the dice. This seems odd; one might expect from the "dice throwing" metaphor that there is a single sheet of graphene oxide deposited on a surface somewhere that the dice may interact with when they are "rolled". Some clarification on this point would be welcome.

My more general critique would be that the purpose of building such nano-dice from DNA and "throwing" them to generate somewhat random numbers is not particularly well motivated. If the goal is to generate stochasticity from a DNA-based nanoscale system, then there would seem to be easier ways to achieve this based on simple competition. Perhaps a significantly expanded Discussion might address this point; as it stands the discussion section is relatively short and largely consists of a summary of the results already presented in the main section of the manuscript.

Reviewer #2 (Remarks to the Author):

Authors reported a very interesting DNA dice with FRET to GO and the data showed statistically reasonable conclusions. This is a high quality study overall and I think it is publishable in Nat. Commun. There are some questions that the authors need to address.

1. The measurements are based on ensemble fluorescence intensity, but dices are counted one at a time. This is a good system to do single fluorescence measurement.
2. Are there any FRET taking place between fluorophores and how does that affect the fluorescence measurement? Authors need to show some fluorescence spectra.
3. Can the dice roll on the GO surface? This would again need single molecule fluorescence measurement.
4. The cheating dice (C) can maximize the probability from 16.7% to 18.5% and cheating dice (T) can minimize the probability 451 from 16.7% to 14.2%. This does not seem to be a very large change.

What's the error of the measurement and are they statistically significant?
5. Is there a way to make the cheating more significant?

Reviewer #3 (Remarks to the Author):

The authors have an interesting intention, which is to simulate dice throwing by adsorption of DNA hexahedron on GO. However, there are many problems in the actual implementation process. I suggest accepting this manuscript only if they address the following concerns.

1. It is important to know clearly the exact pips of the dice to throw. In Scheme 1, the authors labeled four fluorophores on the dice to represent four different signals and indicated by four colors. Are they four different kinds of fluorescence? What are the excitation and emission wavelengths? However, according to the description of method section, it seems to be one. If the same FAM used for F1-4, how to distinguish the 1-6 pips by the fluorescence signal as per the design on scheme 1? This should be clarified.

Furthermore, in Scheme 1, the authors define specific numbering for the different pips, but there is no data to support the specific display of each pip in the experiments.

2. In Fig.1D(ii), the AFM image shows all structures tilted in the same direction, which is unlikely to be the case when structures are randomly adsorbed. Is there something wrong with the imaging? In addition, the authors claimed that "The cube-like nanostructures were observed as a dot in the AFM imaging (Fig. 1D), verifying the successful formation of DNA nano-dice", but, since the outline of the DNA structure cannot be seen and the size of the dots varies widely, does it mean that there are many unformed dice (such as "S1,2,3,4", "S1,2,3,4,x")? The authors did not mention the purification of the structure in the methods section, if the structure formation is incomplete, is the adsorption in GO and the results of fluorescence quantification also questionable?

3. Four nano-dice related dimensions are provided in the article ("side length is of 20 nt (~7 nm)", "average diameter (28.5 ± 2.9 nm)", "theoretical diameter ~20 nm", "theoretical bottom area (max: $\sim 12.56 \times 10^{-18}$ m²)"). In Fig. 1F, what is the diameter of DNA nano-dice refer to and how is it measured? How to calculate the theoretical diameter and theoretical bottom area? Is it a simple geometric calculation or a structural simulation?

4. In the verification of the stochastic system, the authors fixed the number of dice (fixed the concentration) and changed the number of GOs. However, in the actual dice throwing situation, the pips of dice occur randomly on a small number of rolls, and each pip appeared with equal probability after a large number of rolls. Then fixing GO and changing dice seems to be more consistent with the actual throwing situation.

5. The authors did statistical validation of a large number of throwing events, but did not give any proof for individual randomness. Single-molecule studies might be able to support this.

6. Page 8, line 129, the authors claimed that "Due to the adsorption of fluorophore labeled oligonucleotides on GO surface depends on DNA length and base difference", however, there was no significant effect of base differences from the results of Group1 and Group2 in Figure 2.

7. In Supplementary Fig. 1B, the migration rate of dice' is slightly faster than that of dice, why?

8. The schematics of multiple Figures show four simultaneous fluorescence labels of F1-F4 on a dice. If F1-F4 is the same fluorescence, how did the author distinguish the specific one in the fluorescence quantitative analysis? Especially in the flipping model, if the authors label four identical fluorescences, the fluorescence signal may not change even if a c-DNA is added to desorption one t-DNA. If F1-F4 are different fluorescence, please provide the spectral data of each fluorescence before and after flipping.

9. On line 309 in page 17, fluorescence "F" stand for F2, F3, or F1-4? In addition, the structure designed in the author's schematic (Fig.4G) seems to be different from the one used in the actual experiment (Fig.4H). The authors should provide more data to support their design.

10. The authors claimed that "though modulation of the length-dependent bind affinity of ssDNA is also a potential available way for cheating, considering that it would change the overall appearance of the nano-dice, which would be easily debunked and lead to the failure of the cheating, it is not adopted here" in page 19. Actually, in the experiment, the author judged the change of points by the change of fluorescence rather than the appearance of dice, then how could this way be debunked by the fluorescence judgment?

11. On page 20, the author compared the adsorption capacity of polyN12 (N=A, T, C or G) and observed the probability change of "6" by replacing t-DNA-1 and t-DNA-2 with polyN12(N=A, T, C or G). We would suggest the authors to add the sequences in Group 1 and 2 (Fig.2) as control.

12. Several errors should be corrected in the manuscript as following:

(1) "pi-pi stacking" on line 52 in page 3 be unified with " π - π stacking" in the Scheme 1C and on line 345;

(2) "C-DNA-3" should be "c-DNA-3" on line 276.

Response to reviewers

Reviewer 1:

This manuscript describes the use of a DNA cube as a "die" that can be "thrown" by incubating it with graphene oxide sheets; the cubes are adsorbed onto the sheets and the directionality is determined by the quenching (or otherwise) of fluorescent dyes conjugated to the cube via single-stranded DNA oligos. This allows the distribution of die orientations relative to the graphene oxide to be calculated. Furthermore, the manuscript shows that various strategies can be used to bias the die, or "cheat", including by adding oligonucleotides complementary to the dye-carrying protruding strands and by modulating the backbone chemistry of those strands. Both of these approaches essentially weaken the interactions between those strands and the graphene oxide; these need to be balanced carefully to make a die that is "fair".

Response: Many thanks for the valuable comments.

1. The paper is fairly well written, although it is odd that the text continually refers to the structure as a hexahedron rather than a cube: hexahedron is a more general term and it seems that the more specific one would be more appropriate here.

Response: Thanks for the helpful comment. As the reviewer suggested, we have replaced the "hexahedron" with "cube", and corrected the corresponding description in the revised manuscript.

2. The data demonstrating the formation of the cubes is somewhat hard to follow; the figures are rather small and hard to make out (especially the AFM imagery). Also I found it somewhat surprising that all of the data on the behaviors of the dice are in the form of bulk fluorescence readings; given that the "roll" of each individual die is (or at least should be) an independent stochastic event, I would expect to see at least some fluorescence microscopy data here to if nothing else validate that the bulk fluorescence values are at least representative of what one would see if one were to inspect the individual nano-dice one at a time and count up the observed scores from each.

Response: Thank you for thoughtful suggestions. In terms of data for cube formation characterization, we are sorry that AFM imaging has an issue of nanostructure aggregation and disaggregation, which are affected by sample/substrate interactions and sample preparation (e.g. dilution, drying). The scanning results of other references are similar, like Fig. S7 in *J. Am. Chem. Soc.* 2019;141:1100-1108 and Fig. 4d in *Nat. Protoc.* 2020;15:2728-2757. As such, we would like to emphasize that we use AFM or TEM as a rough guide to provide supporting evidence for solution-phase characterization techniques on determining the structures of our nanostructures, and we believe that the most reliable technique that could reflect the molecular structure is agarose gel electrophoresis.

In addition, as suggested, we have monitored corresponding behaviors of the dice using confocal laser scanning microscopy (CLSM) at the whole level and total internal reflection fluorescence microscopy (TIRF) at the single-molecule level. Detailed data and corresponding descriptions have been supplied in the revised manuscript as well as the supplementary information (Line 241-255, Page 14; Line 316-321, Page 18; Fig. 3 and Supplementary Fig. 7).

3. I also wonder what does it mean to "change the concentration of graphene oxide"? The main text

is not especially clear on the details of this protocol but it seems to me that graphene oxide sheets are incubated in the solution freely diffusing with the dice; this is inferred from the discussion that you can get two sheets binding to a single die if there is an excess of graphene oxide with respect to the dice. This seems odd; one might expect from the "dice throwing" metaphor that there is a single sheet of graphene oxide deposited on a surface somewhere that the dice may interact with when they are "rolled". Some clarification on this point would be welcome.

Response: Thank you for valuable comment. "change the concentration of graphene oxide" means that the dice/GO ratio would be adjusted to achieve the purpose of randomness simulation (namely the fluorescence was quenched to half of the original). The corresponding description can be found in the revised manuscript "On this assumption, we optimized the work ratio of GO and DNA dice in order to achieve the purpose of randomly throwing dice with equal probability events. In experiments, we fixed the concentration of DNA dice (500 nM), and adjusted the dice/GO ratio by changing the concentration of GO."(Line 140, Page 8; Line 184, Page 10). We paid more attention to the dice/GO ratio to describe the relationship between dice and GO. In view of the statement that the concentration of GO is excess in the manuscript, our purpose is more to explain the possible reasons for this situation than to associate it with the rolling dice in reality. We have corrected it to a more understandable description in the revised manuscript (Line 139-150, Page 8-9).

4. My more general critique would be that the purpose of building such nano-dice from DNA and "throwing" them to generate somewhat random numbers is not particularly well motivated. If the goal is to generate stochasticity from a DNA-based nanoscale system, then there would seem to be easier ways to achieve this based on simple competition. Perhaps a significantly expanded Discussion might address this point; as it stands the discussion section is relatively short and largely consists of a summary of the results already presented in the main section of the manuscript.

Response: Many thanks again. As the reviewer suggested, we have summarized the results, and further discussed the advantageous feature and breakthrough of our system. Moreover, we have also analyzed the existing challenges and countermeasures one by one. Corresponding descriptions have been supplied in "Discussion" in the revised manuscript (Line 472-550, Page 26-29).

Finally, we would like to express our sincere gratitude. Thank you very much for your valuable comments on this work.

Reviewer 2:

Authors reported a very interesting DNA dice with FRET to GO and the data showed statistically reasonable conclusions. This is a high quality study overall and I think it is publishable in Nat. Commun. There are some questions that the authors need to address.

Response: Many thanks for the valuable comments.

1. The measurements are based on ensemble fluorescence intensity, but dices are counted one at a time. This is a good system to do single fluorescence measurement.

Response: Many thanks for the helpful comments. As the reviewer suggested, we have added corresponding results at the single-molecule level using total internal reflection fluorescence microscopy (TIRF). Detailed data and descriptions have been supplied in the revised manuscript as

well as the supplementary information (Line 241-255, Page 14; Line 316-321, Page 18; Fig. 3 and Supplementary Fig. 7).

2. Are there any FRET taking place between fluorophores and how does that affect the fluorescence measurement? Authors need to show some fluorescence spectra.

Response: Many thanks again. In our experiments, four signals were labeled with FAM fluorophore in turn, so we carried out verification in four batches (F1 to F4) in parallel, that is, when F1 signal was detected, F1 was labeled with FAM, and F2 to F4 were not modified. Thus, there was no interference between spectra. We have added a few words to explain corresponding descriptions in the revised manuscript and provided related spectral raw data in the supplementary information and source data (Line 73-76, Page 4; Line 603-606, Page 32; Supplementary Fig. 2).

3. Can the dice roll on the GO surface? This would again need single molecule fluorescence measurement.

Response: Thanks for your valuable comments. As is suggested, we have supplemented the single-molecule experiments of dice rolling on GO, and the results intuitively showed that the random average distribution of a single fluorescent molecule (dice) on GO would be completely quenched or recovered after being manipulated and rolled, that is, the throwing probability of a “pip” on dice changed from random to unique certainty. Detailed data and corresponding descriptions have been supplied in the revised manuscript as well as the supplementary information (Line 241-255, Page 14; Line 316-321, Page 18; Fig. 3 and Supplementary Fig. 7).

4. The cheating dice (C) can maximize the probability from 16.7% to 18.5% and cheating dice (T) can minimize the probability 451 from 16.7% to 14.2%. This does not seem to be a very large change. What’s the error of the measurement and are they statistically significant?

Response: Many thanks for the helpful comments. The absolute probability of “6” (pip) with different cheating dices was calculated by the following formula: $P_{pip6} = 1/6 \times E_{F3,4}$, which $E_{F3,4}$ means average of normalized F3 and F4, so the error of the measurement was caused by the detected fluorescence. The description of F3 and F4 signals was provided in the Supplementary Table 6. As the reviewer suggested, we further compared the throwing probability of cheating dices with that of normal dice, and analyzed the statistical difference. The results have been added in Fig. 6J.

5. Is there a way to make the cheating more significant?

Response: Thank you for your valuable and thoughtful comment. We want to emphasize that due to the limited length (12 nt) in the t-DNA of dice, the modification on t-DNA would be restricted accordingly, thereby being impossible to achieve complete cheating. At the present stage, the relevant cheating strategies with higher adsorption capacity of GO suggested in the references were basically tried in the manuscript. Considering the simulation of real cheating events, excessive cheating is easy to be seen through. As is suggested, we have also added some examples that might make the cheating more significant in the “Discussion” section. On the other hand, Cheating to the extreme can also be understood as artificial manipulation (that is, directly flipping the dice to the desired pip). We have discussed it in the “Artificial manipulation of DNA nano-dice system” section, which we can achieve the desired pip by adding dual regulators. Corresponding descriptions have been supplied in the revised manuscript (Line 515-541, Page 28-29).

Finally, we would like to express our sincere gratitude. Thank you very much for your valuable comments on this work.

Reviewer 3:

The authors have an interesting intention, which is to simulate dice throwing by adsorption of DNA hexahedron on GO. However, there are many problems in the actual implementation process. I suggest accepting this manuscript only if they address the following concerns.

Response: Many thanks for the valuable comments.

1. It is important to know clearly the exact pips of the dice to throw. In Scheme 1, the authors labeled four fluorophores on the dice to represent four different signals and indicated by four colors. Are they four different kinds of fluorescence? What are the excitation and emission wavelengths? However, according to the description of method section, it seems to be one. If the same FAM used for F1-4, how to distinguish the 1-6 pips by the fluorescence signal as per the design on scheme 1? This should be clarified.

Furthermore, in Scheme 1, the authors define specific numbering for the different pips, but there is no data to support the specific display of each pip in the experiments.

Response: Many thanks for the helpful comments. We have tried to label with four different fluorophores for simultaneous characterization in the past, but unfortunately the result was not ideal, that is, the fluorescence change cannot correspond to the probability. It is speculated that it might be due to the adsorption of fluorophore and GO and other factors. Among these fluorophores, we found that FAM had the weakest effect, so FAM was selected as a unique label in our experiments. By changing the strategy, four signals were labeled with FAM fluorophore in turn, and we carried out verification in four batches (F1 to F4) in parallel, that is, when F1 signal was detected, F1 was labeled with FAM, and F2 to F4 were not modified. We have added a few words to explain corresponding descriptions in the revised manuscript and provided source data (Line 73-76, Page 4; Line 603-606, Page 32; Supplementary Fig. 2). Since we use FAM to characterize the four signals respectively, we can only explain the probability from the perspective of group effect, and cannot independently describe each pip. However, through artificial manipulation, we can adjust the increase/decrease of signals (F1 to F4) to determine the corresponding pips. Therefore, after the addition of dual regulators, we can see from Fig. 5C that only two signals appeared fluorescence, meaning the certain pip.

Furthermore, we have supplemented the fluorescence imaging characterization of dice rolling with dual regulators on GO in Supplementary Fig. 7, which were observed the whole level by confocal laser scanning microscopy (CLSM) and single-molecule level by total internal reflection fluorescence microscopy (TIRF). From this point of view, specific numbering for the different pips could be defined after artificial manipulation (different fluorescence signals of each pip in Fig. 5C and Supplementary Fig. 7). We have added a few words to explain corresponding descriptions in the revised manuscript (Line 241-255, Page 14; Line 316-321, Page 18; Fig. 3 and Supplementary Fig. 7).

2. In Fig.1D(ii), the AFM image shows all structures tilted in the same direction, which is unlikely to be the case when structures are randomly adsorbed. Is there something wrong with the imaging?

In addition, the authors claimed that “The cube-like nanostructures were observed as a dot in the AFM imaging (Fig. 1D), verifying the successful formation of DNA nano-dice”, but, since the outline of the DNA structure cannot be seen and the size of the dots varies widely, does it mean that there are many unformed dice (such as “S1,2,3,4”, “S1,2,3,4,x”)? The authors did not mention the purification of the structure in the methods section, if the structure formation is incomplete, is the adsorption in GO and the results of fluorescence quantification also questionable?

Response: Many thanks again. AFM imaging involves the deposition and drying of DNA solution on another substrate (mica), and the scanning by physical probes might also affect results, which it was possible to cause deformation. The scanning results of other references are similar, like Fig. S7 in *J. Am. Chem. Soc.* 2019;141:1100-1108 and Fig. 4d in *Nat. Protoc.* 2020;15:2728-2757. The AFM was used after purification and the reason why the size of the dots varies widely might be affected by the sample/substrate interactions, sample preparation (e.g. dilution, drying) and probe scanning. As such, we would like to emphasize that we use AFM as a rough guide to provide supporting evidence for solution-phase characterization techniques on determining the structures of our nanostructures, and we believe that the most reliable technique that could reflect the molecular structure is agarose gel electrophoresis. Moreover, we are very sorry that we forgot to add the purification step in the “Methods” section. After the assembly of DNA nano-dices, the nanostructures were purified with Amicon centrifugal filter (10 kDa molecular weight cut-off) and then centrifuged at 10000 g for 10 min three times to remove unhybridized ssDNA. Corresponding descriptions have been supplied in the revised manuscript (Line 565, Page 30). Subsequent experiments were conducted after purification and it can be seen in AGE characterization in Fig. 1B that the synthetic product at every stage was a very pure band, meaning no additional intermediates (unformed dices).

3. Four nano-dice related dimensions are provided in the article (“side length is of 20 nt (~7 nm)”, “average diameter (28.5 ± 2.9 nm)”, “theoretical diameter ~20 nm”, “theoretical bottom area (max: ~12.56 × 10⁻¹⁸ m²)”). In Fig. 1F, what is the diameter of DNA nano-dice refer to and how is it measured? How to calculate the theoretical diameter and theoretical bottom area? Is it a simple geometric calculation or a structural simulation?

Response: Many thanks again. The “diameter of DNA nano-dice” in Fig. 1F refers to the diameter distribution of each particle in the TEM imaging. Since the imaging effect is two-dimensional and cannot present a three-dimensional structure, we can only use image-j software to roughly evaluate the diameter. The theoretical diameter of DNA nano-dice is a simple geometric calculation based on the longest diagonal diameter. The t-DNA is 12 nt and the internal diagonal of the cube is radical 2 the length of the side (20 nt). Therefore, the theoretical length should be $(2 \times 12\text{nt} + \sqrt{3} \times 20\text{nt})$. The length of each base is about 0.34 nm, and thus, the theoretical diameter = $(2 \times 12\text{nt} + \sqrt{3} \times 20\text{nt}) \times 0.34 \approx 20$ nm. The theoretical bottom area is estimated from the square of the theoretical diameter and we are very sorry for the mistake for the value, which is incorrect introduction of intermediate calculated values. We have carefully corrected the value to “~39.8 × 10⁻¹⁷ m²” in the revised manuscript (Line 155, Page 9).

4. In the verification of the stochastic system, the authors fixed the number of dice (fixed the concentration) and changed the number of GOs. However, in the actual dice throwing situation, the

pips of dice occur randomly on a small number of rolls, and each pip appeared with equal probability after a large number of rolls. Then fixing GO and changing dice seems to be more consistent with the actual throwing situation.

Response: Many thanks again. Whether adjusting the concentration of GO or dice, our aim is to obtain the appropriate dice/GO ratio in the simulation of random. When fixing the concentration of GOs, a small number of dice rolling means dice/GO ratio $<1.2 \times 10^{-5}$ mol/g (optimal ratio calculated in the manuscript), and a large number of dice rolling means dice/GO $>1.2 \times 10^{-5}$ mol/g. Even though it seems more consistent with the actual throwing situation to fix GO and change dice, the proportion relationship between them is still consistent from the result analysis, and we paid more attention to the dice/GO ratio to describe the relationship between dice and GO. Also, We added some understandable descriptions in the revised manuscript (Line 139-150, Page 8-9).

5. The authors did statistical validation of a large number of throwing events, but did not give any proof for individual randomness. Single-molecule studies might be able to support this.

Response: Many thanks again. As the reviewer suggested, we have added corresponding results at the single-molecule level. Detailed data and descriptions have been supplied in the revised manuscript as well as the supplementary information (Line 241-255, Page 14; Line 316-321, Page 18; Fig. 3 and Supplementary Fig. 7).

6. Page 8, line 129, the authors claimed that “Due to the adsorption of fluorophore labeled oligonucleotides on GO surface depends on DNA length and base difference”, however, there was no significant effect of base differences from the results of Group1 and Group2 in Figure 2.

Response: Many thanks again. The purpose of this sentence is to explain the factors that affect the adsorption of ssDNA and GO, so we further indicate that “we strictly constrained the sequence composition of the four t-DNAs on the nano-dice to make them equivalent when binding GO” in the manuscript (Line 126-129, Page 8). The sequence composition of four t-DNAs in Group 1 and Group 2 was different, but the difference was the overall distinction of the dices. The sequence length and base proportion of t-DNAs used for adsorption with GO in dice and dice' were still the same, which the proportions of A/T/C/G are fixed at 25% in the t-DNA in dice (Group 1) and A/C accounts for 42% and T/G accounts for 8% in the t-DNA in dice' (Group 1). Only the internal differences of the same dice (the sequence composition of four t-DNAs were inconsistent) would cause the difference in throwing, which was similar to the results explored in our cheating part. Therefore, the preparation of dice and dice' is a universal exploration, indicating that stochastic effect can still be simulated. And in dice' (Group 2), due to the change of t-DNA sequence, the GO concentration required for dice' random simulation has also changed to some extent, which is the overall difference between dice and dice'.

7. In Supplementary Fig. 1B, the migration rate of dice' is slightly faster than that of dice, why?

Response: Many thanks again. The difference may be caused by the following reasons. First, the gel was not aligned during shooting, resulting in visual errors, and we provide raw data to support. Second, through investigation, pKa values of cytidine, adenosine, and guanosine were reported to be 4.2, 3.5, and 1.9, respectively (*Outlines of Biochemistry, 5th ed., John Wiley & Sons, Chichester, 1987*), which the order of pH values at which electrophoretic mobility began to decrease corresponds to that of pKa values (*J. Sep. Sci. 2017;40:3153-3160*). However, since the

concentration of our electrophoresis solution does not reach the special acidic condition, we may not be able to accurately identify the difference of its base. We are willing to conduct in-depth research on related issues in the future and present it in the form of another paper. Based on the support of the original data (below, also in source data), we believed that there was almost no difference in the migration rate between dice and dice'.

Figure S1B

8. The schematics of multiple Figures show four simultaneous fluorescence labels of F1-F4 on a dice. If F1-F4 is the same fluorescence, how did the author distinguish the specific one in the fluorescence quantitative analysis? Especially in the flipping model, if the authors label four identical fluorescences, the fluorescence signal may not change even if a c-DNA is added to desorption one t-DNA. If F1-F4 are different fluorescence, please provide the spectral data of each fluorescence before and after flipping.

Response: Many thanks again. For the problem of fluorescent labeling, we have given feedback in question 1 of reviewer. In our experiments, four signals were labeled with FAM fluorophore in turn, so we carried out verification in four batches (F1 to F4) in parallel, that is, when F1 signal was detected, F1 was labeled with FAM, and F2 to F4 were not modified. We have added a few words to explain corresponding descriptions in the revised manuscript (Line 73-76, Page 4; Line 603-606, Page 32). Thus, when adding c-DNA for manipulation, since c-DNA and t-DNA are one-to-one, the fluorescence signal can be detected normally without any impact. We also provided related spectral raw data of each fluorescence before and after in the file "source data". Due to the large amount of data involved, please allow us to provide spectral data at the peak in the source data. If necessary, we can also provide full spectral data later.

9. On line 309 in page 17, fluorescence "F" stand for F2, F3, or F1-4? In addition, the structure designed in the author's schematic (Fig.4G) seems to be different from the one used in the actual experiment (Fig.4H). The authors should provide more data to support their design.

Response: Many thanks again. The fluorescence "F" stand for F2 and F3. We have carefully corrected the description in the revised manuscript (Line 343, Page 19). In addition, the structure designed in Fig. 5G in the revised manuscript (namely Fig. 4G in the original manuscript) is the schematic illustration of c-DNA-2 and cc-DNA-3 adding to DNA nano-dice system with c-DNA-3 and c-DNA-4, while the structure in Fig. 5H in the revised manuscript (namely Fig. 5H in the original manuscript) is theoretical positive control dice for F2 and negative control dice for F3, which were used to normalize the fluorescence signals for F2 and F3, respectively. We also provided

related spectral raw data in the the source data.

10. The authors claimed that “though modulation of the length-dependent bind affinity of ssDNA is also a potential available way for cheating, considering that it would change the overall appearance of the nano-dice, which would be easily debunked and lead to the failure of the cheating, it is not adopted here” in page 19. Actually, in the experiment, the author judged the change of points by the change of fluorescence rather than the appearance of dice, then how could this way be debunked by the fluorescence judgment?

Response: Many thanks again. Cheating is characterized by fluorescence change, showing the change of overall dice probability. And whether the way could be debunked is not judged by the fluorescence. The purpose of this description is to show that if we modify the dice too much for cheating in reality, it will be easy to find. Similarly, when we modify the t-DNA of dice too long, its overall appearance will also be greatly changed, leading to be easily debunked. In view of the ambiguity in this description, we have deleted it in the revised manuscript.

11. On page 20, the author compared the adsorption capacity of polyN₁₂ (N=A, T, C or G) and observed the probability change of "6" by replacing t-DNA-1 and t-DNA-2 with polyN₁₂(N=A, T, C or G). We would suggest the authors to add the sequences in Group 1 and 2 (Fig.2) as control.

Response: Many thanks again. As is suggested, we investigated the dice (group 1) and dice' (group 2) cheating by modulating base composition, respectively. The desorption results of sequence in group 1 and group 2 (Fig. 6C and Supplementary Fig. 14A) were basically consistent and meet the following relationship: polyC₁₂ > polyA₁₂ > polyG₁₂ > polyT₁₂. Moreover, the results of cheating dice and dice' (Fig. 6E and Supplementary Fig. 14B) were also in good accordance with corresponding competitive sequence experimental above. Detailed data and descriptions have been supplied in the revised manuscript as well as the supplementary information (Line 397-399, Page 22; Fig. 6C, E, and Supplementary Fig. 14).

12. Several errors should be corrected in the manuscript as following:

(1) “pi-pi stacking” on line 52 in page 3 be unified with “ π - π stacking” in the Scheme 1C and on line 345;

(2) “C-DNA-3” should be “c-DNA-3” on line 276.

Response: Many thanks again, and we have corrected the corresponding description in the revised manuscript.

Finally, we would like to express our sincere gratitude. Thank you very much for your valuable comments on this work.

REVIEWERS' COMMENTS

Reviewer #1 (Remarks to the Author):

The authors seem to have done a good job addressing the concerns raised by the first set of reviews. In particular, the inclusion of fluorescence microscopy data is an important addition to demonstrate that the DNA nano-dice do behave as implied by the bulk fluorescence results.

A few minor comments:

- In Fig S7, it would be helpful to label the figure somehow to specify which F1-4 channel corresponds to which dye that is expected to be fluorescing on the nanostructure when it is lying on the graphene oxide surface in that orientation.
- As a matter of language, "dices" is not a word (I noticed this appearing at least once in the SI and maybe in the main text though I didn't scan through too closely). The singular is actually "die" and "dice" is already the plural.

Reviewer #2 (Remarks to the Author):

My comments have been adequately addressed. I now recommend publication of this work.

Reviewer #3 (Remarks to the Author):

The authors have properly addressed the reviewers' comments in the revised manuscript and hence I recommend publication of this work.

Response to reviewers

Reviewer #1:

The authors seem to have done a good job addressing the concerns raised by the first set of reviews. In particular, the inclusion of fluorescence microscopy data is an important addition to demonstrate that the DNA nano-dice do behave as implied by the bulk fluorescence results.

Response: We would like to express our sincere gratitude and many thanks for the valuable comments.

A few minor comments:

1. In Fig S7, it would be helpful to label the figure somehow to specify which F1-4 channel corresponds to which dye that is expected to be fluorescing on the nanostructure when it is lying on the graphene oxide surface in that orientation.

Response: Thank you for thoughtful suggestions. As the reviewer suggested, we have labeled the Supplementary Fig. 7 in the revised manuscript.

2. As a matter of language, "dices" is not a word (I noticed this appearing at least once in the SI and maybe in the main text though I didn't scan through too closely). The singular is actually "die" and "dice" is already the plural.

Response: Thanks for the helpful comment. As the reviewer suggested, we have carefully corrected the corresponding description in the revised manuscript and supplementary information.

Reviewer #2:

My comments have been adequately addressed. I now recommend publication of this work.

Response: We would like to express our sincere gratitude and many thanks for the valuable comments.

Reviewer #3:

The authors have properly addressed the reviewers' comments in the revised manuscript and hence I recommend publication of this work.

Response: We would like to express our sincere gratitude and many thanks for the valuable comments.

Finally, we would like to express our sincere gratitude. Thank you very much for your valuable comments on this work.